JCB Journal of Cell Biology

# Disruption of macrophage cell volume drives inflammatory responses and type I interferon signaling

James R. Cook[1,6], Tara A. Gleeson[1,2,3], Sara Gago[4], Stuart M. Allan[1,2,3], Kevin N. Couper[2,3,5], Catherine B. Lawrence[1,2,3], David Brough[1,2,3], and Jack P. Green[1,2,3]

Macrophages coordinate inflammatory and immune responses to threats, yet how they interpret diverse danger signals to tailor inflammation remains unclear. Disturbances in extracellular and intracellular homeostasis alter cell volume, but the consequences for macrophage inflammatory responses are poorly understood. We demonstrate that macrophages use cell volume control as a danger-sensing mechanism to promote and augment inflammation. Using volume-regulated anion channel (VRAC)–deficient macrophages, which lack cell volume control under hypo-osmotic conditions, we show that cell volume disruptions drive transcriptomic reprogramming and induction of inflammation. Cell volume disruption induced type I interferon signaling through a DNA- and TBK1-dependent mechanism, but independent of cGAS and 2′3′-cGAMP transport. VRAC deficiency enhanced macrophage antiviral responses to influenza infection. Cell volume changes synergized with diverse pathogen-associated molecular pattern–mediated signaling to augment type I interferon responses and exacerbate the cytokine storm in mouse models of hyperinflammation. Our findings highlight cell volume as an important regulator in shaping inflammatory responses, expanding our understanding of how macrophages sense complex danger signals.

## Introduction

The innate immune system is an early line of defense against invading pathogens and tissue damage and functions to promote pathogen clearance and wound healing. Macrophages represent the cellular frontline of the innate immune system and are specialized to detect danger signals such as features of infection (pathogen-associated molecular patterns, PAMPs), cellular damage (damage-associated molecular patterns, DAMPs), or disturbances in extracellular and intracellular homeostasis, to initiate and sustain inflammatory responses (Li and Wu, 2021). The repertoire of danger signals sensed by macrophages is diverse and extensive highlighting a crucial role of macrophages in danger sensing (Ley et al., 2016; Wang and Labzin, 2023). Furthermore, the nature of the inflammatory response initiated varies depending on the type of danger signal sensed to resolve the threat most appropriately. The mechanisms utilized by immune cells to decipher the many different danger signals in the inflammatory environment to promote an optimal response are an area with many unanswered questions.

A relatively unexplored contributor to the tuning of inflammatory responses is the regulation of macrophage cell volume. Cell volume control is critical for normal cellular function, where excessive changes in cell volume threaten cell survival and homeostasis. Cell volume is constantly regulated by proliferative and metabolic processes where loss of cell volume control, and resulting cell volume dysfunction, is commonly associated with features of the inflammatory environment, such as edema (Luo et al., 2020), hypoxia, and metabolic dysfunction (Osei-Owusu et al., 2018), with cell swelling a common feature of pro-inflammatory cell death (Okada et al., 2020). Cells use mechanisms to regulate cell volume, including ion channels such as the volume-regulated anion channel (VRAC) in vertebrates. In swollen cells, VRAC drives chloride ion (Cl$^-$) and osmolyte efflux to promote water efflux to restore cell volume, through a process known as regulatory volume decrease (RVD) (Qiu et al., 2014; Voss et al., 2014). Loss of VRAC results in dysregulated cell swelling following hypotonic shock through an inability to undergo RVD (Green et al., 2020; Voss et al., 2014). Since the discovery of LRRC8 family proteins as the subunits of VRAC (Qiu et al., 2014; Voss et al., 2014), it is now known that as well as cell volume regulation, VRAC is implicated in diverse cellular functions such as cell migration, proliferation, apoptosis, cell–cell

[1]Division of Neuroscience, School of Biological Sciences, Faculty of Biology, Medicine and Health, University of Manchester, Manchester, UK; [2]Geoffrey Jefferson Brain Research Centre, The Manchester Academic Health Science Centre, Northern Care Alliance NHS Foundation Trust, University of Manchester, Manchester, UK; [3]Lydia Becker Institute of Immunology and Inflammation, University of Manchester, Manchester, UK; [4]Manchester Fungal Infection Group, Division of Evolution, Infection and Genomics, Faculty of Biology, Medicine and Health, University of Manchester, Manchester, UK; [5]Division of Immunology, Immunity to Infection and Respiratory Medicine, Faculty of Biology, Medicine & Health, University of Manchester, Manchester, UK; [6]Institute of Neuropathology, Medical Faculty, University of Freiburg, Freiburg, Germany.

Correspondence to Jack P. Green: jack.green@manchester.ac.uk.

communication, and immunity (Chen et al., 2019). VRAC and the RVD are emerging as important regulators of inflammatory signaling in immune cells by regulating activation of the NLRP3 inflammasome (Chirayath et al., 2024; Green et al., 2020; Wu et al., 2023), and cyclic GMP-AMP synthase (cGAS)–stimulator of interferon genes (STING) pathways (Chen et al., 2021; Lahey et al., 2020; Zhou et al., 2020a). Furthermore, treatment with hypertonic solutions, which prevents cell swelling, is effective at alleviating inflammation *in vivo*, including in LPS-induced sepsis (Theobaldo et al., 2012), intracerebral hemorrhage (Schreibman et al., 2018), pulmonary injury due to ischemia–reperfusion (Shields et al., 2003), and acute respiratory distress syndrome (ARDS) (Petroni et al., 2015). However, despite the reported indications that cell volume and VRAC are involved in inflammatory signaling, the basic biological mechanisms of how the regulation of cell volume shapes inflammation are currently unknown.

Here, we demonstrate that regulation of macrophage cell volume shapes the innate immune response. By using conditional VRAC knockout (KO) macrophages, we revealed that loss of cell volume control resulted in an altered transcriptional response to hypotonicity, including an induction of inflammatory signaling promoting an antiviral cell state. We show that swelling of macrophages drove type I interferon signaling through a DNA- and TBK1-dependent pathway identifying a new mechanism of inflammation following cell swelling. We show that VRAC-deficient macrophages exhibited altered cytokine responses following stimulation of PAMPs in hypo-osmotic conditions, promoting enhanced type I IFN responses, suggesting that regulation of cell volume influences the nature of the immune response. Furthermore, we provide evidence that the regulation of macrophage cell volume contributes to inflammation in disease, where VRAC-deficient macrophages exhibited increased antiviral responses to influenza A infection, and finally *in vivo* by observing exacerbated inflammatory responses in conditional CX3CR1 cell-specific VRAC KO mice in a model of CpG-DNA–induced hyperinflammation. Therefore, we propose that macrophage cell volume control is critical for shaping inflammatory responses.

## Results

### Altered regulation of macrophage cell volume initiates transcriptomic reprogramming and induction of inflammation

We hypothesized that perturbations in the ability of a cell to control its cell volume would influence inflammatory signaling. We have previously described and characterized macrophages from mice lacking LRRC8A (a subunit essential to all VRAC channels [Qiu et al., 2014; Voss et al., 2014]) in CX3CR1+ve cells (*Cx3cr1*Cre *Lrrc8a*fl/fl mice, termed throughout as VRAC KO macrophages), which lack the ability to undergo RVD in response to hypotonic shock, and as a result stay swollen (Cook et al., 2022; Green et al., 2020). Therefore, VRAC KO macrophages are an ideal tool to investigate how a loss of cell volume control regulates inflammatory signaling. We have previously shown that upon exposure to severe hypotonic conditions (117 mOsm kg−1), that the macrophage RVD response induces VRAC-dependent activation of the NLRP3 inflammasome (Green et al., 2020),

providing evidence that control of cell volume influences inflammatory signaling. However, these are extreme hypotonic conditions, leading us to question how macrophages would respond to less severe changes in cell volume in circumstances where the inflammasome is not activated. To this end, we first performed bulk RNA sequencing (RNA-seq) on wild-type (WT) and VRAC KO macrophages following less severe cell volume disruption to assess what, if any, transcriptomic changes occurred. Primary bone marrow–derived macrophages (BMDMs) generated from WT and VRAC KO littermate mice were treated with hypo-osmotic media (50% vol/vol $H_2O$ in DMEM, ~170 mOsm kg−1) to induce cell volume changes or left untreated (Fig. 1 A). Cre control (*Cx3cr1*cre *Lrrc8a*WT/WT) macrophages were also included in the untreated group (Fig. 1 A). In parallel, we confirmed that the VRAC KO macrophage cultures used for RNA-seq had lost the ability to undergo RVD in response to hypo-osmotic media (Fig. S1, A and B).

Principal component analysis indicated very little variability among the WT, VRAC KO, and Cre control macrophages under isotonic conditions, whereas hypotonic-stimulated macrophages clustered separately, with further separation between the VRAC KO and WT cells along PC1 (Fig. 1 B). Differential expression analysis similarly detected very few differentially expressed genes (DEGs) between genotypes under isotonic conditions (2, 4, and 6 DEGs for the VRAC KO vs WT, Cre vs WT, and VRAC KO vs Cre comparisons, respectively), but with wide-ranging changes induced by hypotonic swelling, resulting in 6,315 DEGs in WT cells (Fig. 1 C) and 7,171 DEGs in KO cells (Fig. 1 D) relative to their respective isotonic controls. In contrast to isotonic conditions, we further detected 3,563 DEGs between VRAC KO and WT macrophages under hypotonic conditions (Fig. 1 E), indicating that loss of volume control significantly altered the cellular response to hypotonic swelling.

We performed gene set enrichment analysis (GSEA) (Subramanian et al., 2005) to first determine which pathway-level transcriptional responses could be identified in WT cells following hypotonic swelling. We used the Molecular Signatures Database (MSigDB) to query Gene Ontology, MSigDB Hallmark, Wiki-Pathways, and KEGG gene sets, summarizing the results using an enrichment map to group redundant/overlapping terms into clusters representing discrete biological themes (Fig. 1 F). Gene set analysis reports of upregulated and downregulated gene sets are shown in Table S1. We observed a marked downregulation of cholesterol synthesis, mRNA processing, and DNA replication pathways under hypotonic conditions, along with upregulation of many gene sets relating to inflammation and immune responses, in addition to a cluster of terms representing morphological and developmental processes. The latter group may represent a significant number of cytoskeletal and cell adhesion genes regulated by hypotonicity, as well as Wnt and Notch pathway members (e.g., Wnt4, Dll1). The most significantly upregulated gene sets relating to inflammation included the hallmark interferon alpha and gamma, TNF, and NF-κB response pathways, indicating a wide-ranging upregulation of inflammatory genes in response to cell volume disruption.

We next investigated how the response to hypotonic stimulation was altered by loss of VRAC. VRAC KO macrophages

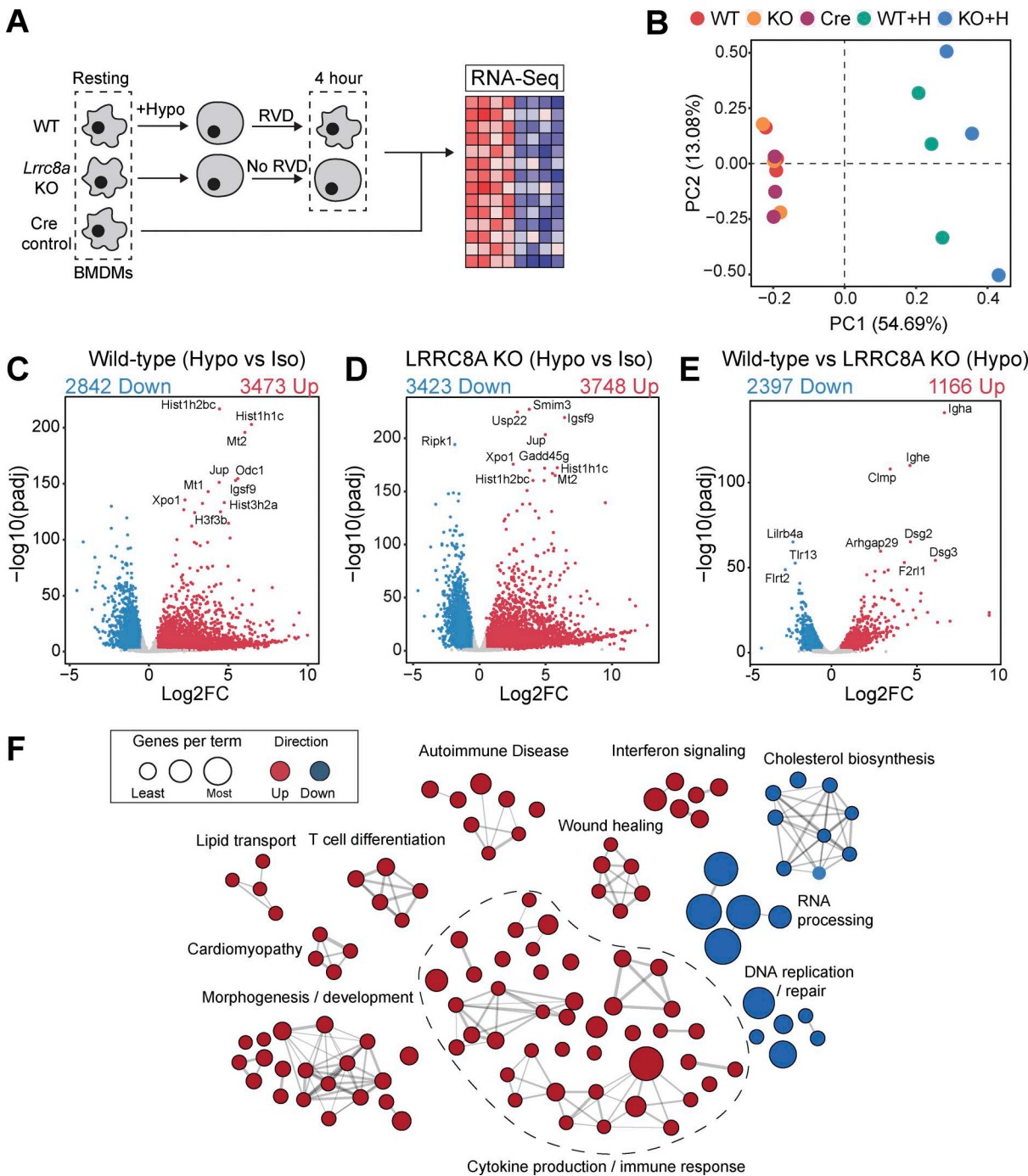

Figure 1. **Changes in macrophage cell volume induce a large transcriptomic response that alters diverse cellular processes. (A and B)** RNA-seq was performed on resting BMDMs from WT, VRAC KO, or CX3CR1 Cre–expressing controls (Cre), and WT and KO BMDMs incubated in hypo-osmotic media (50% vol/vol H₂O in DMEM) for 4 h (n = 3/condition). (B) Principal component analysis of the gene-wise read counts from A. **(C–E)** Volcano plots of differentially regulated genes (DEGs) in: resting WT macrophages vs WT in hypo-osmotic media (C), resting KO macrophages vs KO in hypo-osmotic media (D), WT macrophages in hypo-osmotic media vs KO macrophages in hypo-osmotic media (E). **(F)** Enrichment map of cellular processes differentially regulated between WT and WT treated with hypo-osmotic media.

stimulated with hypotonicity displayed a response, which over-lapped largely with that of WT macrophages, although a significant number of genes were induced only in VRAC KO cells (1,781), as well as some DEGs that were induced only in WT (642) (Fig. S1 C). VRAC also affected the expression pattern of many shared DEGs, with 1,435 genes displaying an enhanced regulation in VRAC KO (that is, a significant differential expression between KO and WT cells under hypotonic conditions in the same direction as the regulation in WTs versus isotonic conditions), and 701 genes displaying suppressed regulation, indicating that the presence or absence of VRAC modulates the hypotonic response to a significant extent (Fig. S1, D and E). GSEA revealed that the most

pronounced pathway-level differences between the VRAC KO and WT responses involved cytoskeletal and adhesion molecules (e.g., Dsg2/3, Dsc1/2/3) (Fig. S1 F), whose upregulation was greatly enhanced in the absence of VRAC. Additionally, there was a milder downregulation of the cholesterol synthesis pathway (Fig. S1 G), and a dramatic suppression of mRNA processing genes (Fig. S1 H) was observed in VRAC KO macrophages. Thus, failure to undergo RVD and the associated sustained cell swelling strongly modifies the transcriptomic response both by enhancing and by suppressing the induction of diverse responses. Collectively, these findings demonstrate that cell volume disruption in macrophages triggers transcriptional reprogramming affecting diverse cellular functions and activates inflammatory signaling pathways.

## Cell swelling induces an antiviral response through type I IFN signaling

Since we hypothesized that perturbations in cell volume would drive inflammatory responses in macrophages, we examined the expression of cytokines within the hypo-osmotic treatment RNA-seq dataset in more detail. Hypo-osmotic cell swelling induced a widespread induction of multiple cytokines, including IL-1α, IL-1β, IL-6, IL-33, TNF, CCL2, and CD40 among others (Fig. 2 A). We observed that hypo-osmotic cell swelling caused an induction of the type I IFNs, particularly IFNβ, but not type II IFN (IFNγ) or type III IFNs (IFNλ) (Fig. 2 B). Further, while type I IFN expression was mostly absent at baseline in WT and VRAC KO BMDMs, VRAC KO BMDMs exhibited a larger induction of type I IFNs than WT BMDMs following hypo-osmotic treatment (Fig. 2 B). Treatment with hypo-osmotic media also induced expression of interferon-stimulated genes (ISGs), which are commonly transcribed downstream of IFN signaling (Schoggins and Rice, 2011), although there was no difference in ISG upregulation between WT and VRAC KO macrophages following 4 h of incubation in hypo-osmotic media (Fig. 2 C). We then examined the kinetics of type I IFN signaling in WT and VRAC KO BMDMs using qRT-PCR for *Ifnb* and the ISG *Cxcl10*. While there was no noticeable upregulation of *Ifnb* in WT BMDMs following treatment with hypo-osmotic media, an upregulation of the ISG *Cxcl10* was present at 4 h and enhanced further at 6 h (Fig. 2, D and E). Interestingly in VRAC KO BMDMs, *Ifnb* levels were upregulated following 4 h of hypo-osmotic treatment, which was further augmented at 6 h (Fig. 2 D). *Cxcl10* was also upregulated from 4 h in hypo-osmotic medium–treated VRAC KO BMDMs, which was significantly higher than that of WT BMDMs at 6 h (Fig. 2 E). The expression of *Ifna*, the ISGs *Rsad2* and *Ccl7*, and the inflammatory cytokines *Il1b*, *Il18*, and *Tnf* was also upregulated at 6 h following hypotonicity-induced cell swelling in WT and VRAC KO BMDMs, with a significantly enhanced response seen in VRAC KO BMDMs (Fig. 2, F and H; and Fig. S2, A–C). Together, these data suggest that enhanced IFNβ signaling at earlier time points could be promoting greater ISG expression at later time points. While upregulated mRNA expression of *Ifnb* occurred following treatment with hypo-osmotic media, IFNβ was not detected extracellularly, potentially due to the low sensitivity of the ELISA, suggesting hypo-osmotic media induced only a mild interferon response compared with stimuli such as nucleic acid addition or

viral infection (data not shown). However, the induction of viperin (translated from *Rsad2)* was detectable from 4 h and was slightly, but significantly, enhanced in VRAC KO BMDMs at 8 h (Fig. 2, I and J) matching mRNA data at 6 h (Fig. 2 F). We then used BMDMs isolated from interferon-α/β receptor (IFNAR) KO mice (Muller et al., 1994), which are unable to respond to type I IFNs, to examine the importance of type I IFN signaling on the induction of inflammatory responses by hypo-osmotic media. IFNAR KO BMDMs did not induce the ISG viperin in response to recombinant IFNβ or poly I:C demonstrating their insensitivity to type I IFNs (Fig. S2, D and E). IFNAR KO BMDMs also lacked the induction of the ISGs *Cxcl10* and *Rsad2* (viperin) by hypo-osmotic media, which was still present in WT littermate controls (Fig. 2, K–M). Conversely, the induction of the NF-κB–driven cytokine pro-IL-1β was unaltered between WT and IFNAR KO BMDMs (Fig. 2 M), suggesting type I IFN was not upstream of all inflammatory responses induced by hypo-osmotic media. These data demonstrate that changes in macrophage cell volume by hypo-osmotic shock cause inflammation, particularly type I IFN signaling, which is enhanced when cells are unable to undergo RVD.

We then examined potential mechanisms of type I IFN signaling induction following cell volume changes. We performed Ingenuity Pathway Analysis on our RNA-seq data from hypo-osmotic medium–challenged WT BMDMs to predict which kinases and transcription factors could be orchestrating this response (Fig. 2 N). The 10 transcription factors and kinases with the highest and lowest Z-scores are shown in Fig. 2, O and P. Interestingly, the regulators most likely to be activated following changes in cell volume by hypo-osmotic media were core components of antiviral signaling pathways, such as IRF3, IRF7, STAT1, and MAP3K7 (TAK1), and of pro-inflammatory signaling through NF-κB, such as RELA, HIF1A, IKBKB, IKBKG, and CHUK (IKKα) (Fig. 2, O and P). Correspondingly, antagonistic pathways of antiviral signaling were predicted to be downregulated, such as TRIM24. Further, there was predicted activation of sensors of cytosolic nucleotides, in particular TLR3, STING, TLR9, TLR7, MAVS, cGAS, and IFIH1, which can all function upstream of interferon regulatory factor (IRF) signaling to drive type I IFN induction (Fig. 2 Q). These analyses suggested that changes in cell volume promote activation of nucleic acid–sensing pathways leading to a type I IFN response and induction of an inflammatory cell state.

We next sought to identify the cellular mechanisms behind cell swelling induction of type I IFNs. TBK1 is a central common regulator downstream of diverse nucleic acid–sensing pathways (Zhou et al., 2020b). Treatment of WT BMDMs with the TBK1/IKKε inhibitor MRT67307 blocked the hypo-osmotic induction of viperin (Fig. 3, A and B), demonstrating that type I IFN is dependent on TBK1/IKKε signaling. We ruled out endosomal Toll-like receptors (TLRs) using bafilomycin A1 (BafA1), a V-ATPase inhibitor that causes endolysosomal alkalinization and impairs TLR3, TLR7, TLR8, and TLR9 responses (Blasius and Beutler, 2010). BafA1 blocked poly I:C-induction of viperin in WT BMDMs, while hypo-osmotic induction of viperin was unaffected (Fig. 3, C and D). We next explored involvement of STING, an inducer of type I IFNs in response to double-stranded

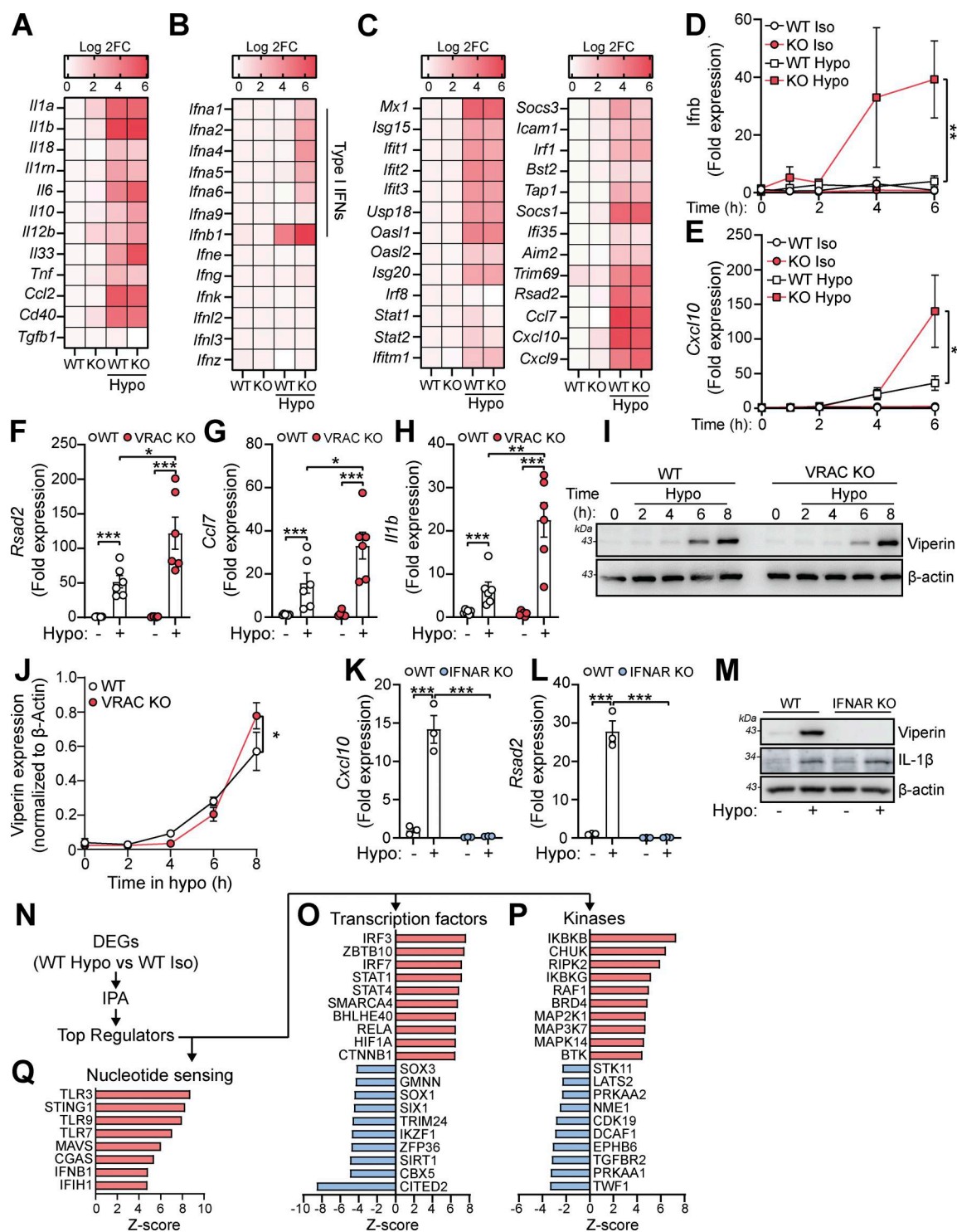

**Figure 2. Changes in macrophage cell volume promote inflammation through type I Interferon signaling. (A–C)** Heatmaps of mRNA expression of cytokines (A), interferons (B), and ISGs (C) from RNA-seq of WT or VRAC KO BMDMs ± incubation in hypo-osmotic media (50% vol/vol $H_2O$ in DMEM). **(D and E)** qRT-PCR analysis of *Ifnb* and *Cxcl10* in WT and VRAC KO BMDMs ± incubation in hypo-osmotic media for the indicated time points ($n = 4$). **(F–H)** qRT-PCR analysis of *Rsad2* (F), *Ccl7* (G), and *Il1b* (H) in WT and VRAC KO BMDMs ± incubation in hypo-osmotic media for 6 h ($n = 6$). **(I)** Western blot for viperin in WT and VRAC KO BMDMs incubated in hypo-osmotic media for the indicated time points ($n = 4$). **(J)** Densitometry of (I) ($n = 4$). **(K and L)** qRT-PCR analysis of *Cxcl10* (K) and *Rsad2* (L) in WT and IFNAR KO BMDMs ± incubation in hypo-osmotic media (6 h) ($n = 3$). **(M)** Western blot for viperin and pro-IL-1β in WT and IFNAR KO BMDMs incubated in hypo-osmotic media (6 h) ($n = 3$). **(N)** Pipeline for IPA of RNA-seq data (from Fig. 1 A) to identify potential differential upstream regulators in WT BMDMs following incubation in hypo-osmotic media. **(O and P)** Top 10 predicted upregulated and downregulated transcription factors (O) and kinases (P). **(Q)** Predicted upregulated nucleotide-sensing pathways. *$P < 0.05$, **$P < 0.01$, ***$P < 0.001$, determined by a one-way ANOVA (2D and E) or by a two-way ANOVA (2F–L) with Sidak's post hoc analysis. Values shown are the mean ± the SEM. IPA, Ingenuity Pathway Analysis. Source data are available for this figure: SourceData F2.

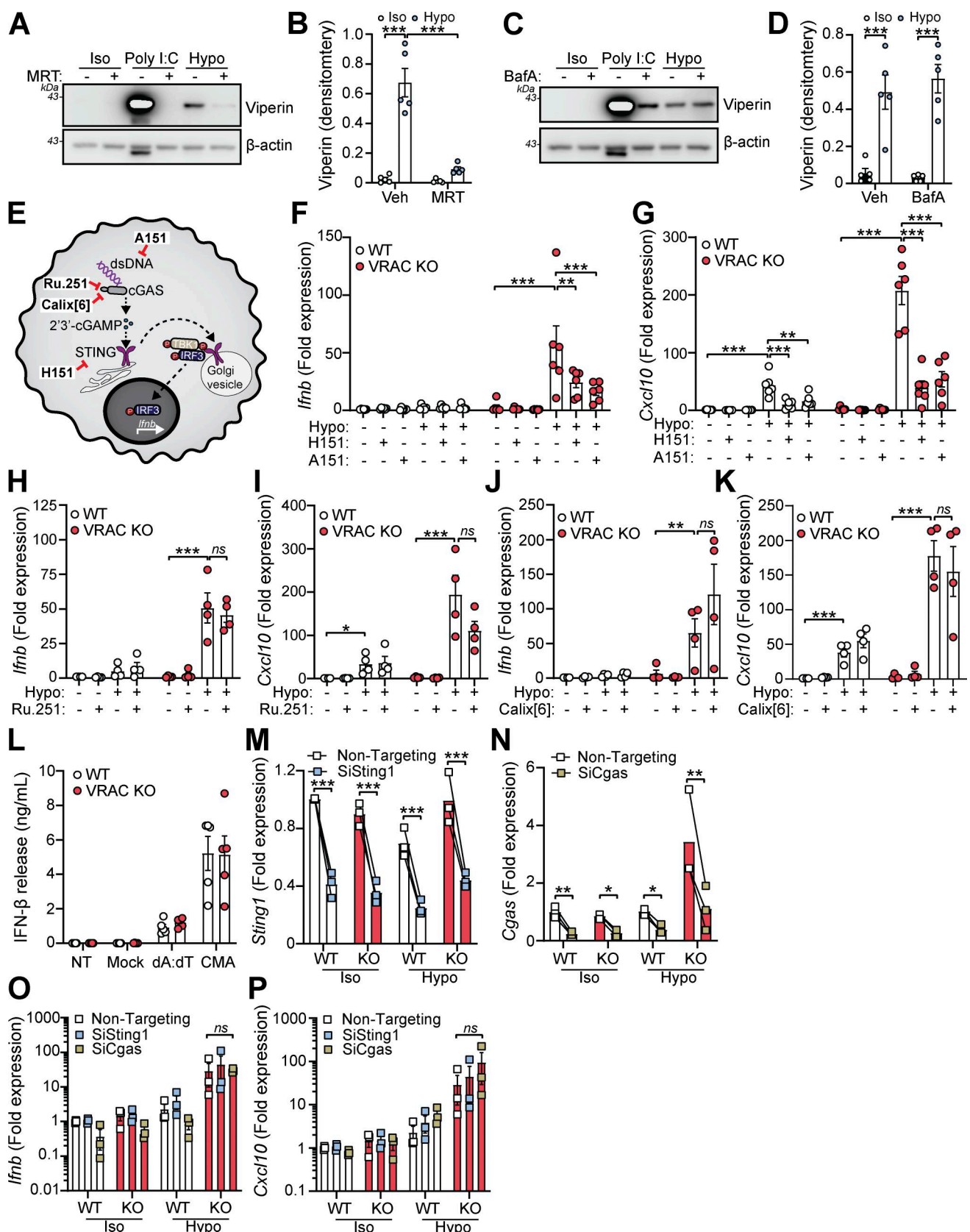

Figure 3. **Changes in cell volume drive IFNβ responses through a DNA- and TBK1-dependent pathway. (A and B)** Western blot (A) and densitometry (B) of viperin in WT BMDMs incubated in iso-osmotic media (± the TLR3 agonist poly I:C [1 µg ml⁻¹]) or in hypo-osmotic media (50% H2O vol/vol in DMEM) in the presence of the TBK1/IKKε inhibitor MRT67307 (MRT, 10 µM) or vehicle control (DMSO, 0.1% vol/vol) (6 h) (n = 5). **(C and D)** Western blot (C) and densitometry (D) of viperin in WT BMDMs incubated in hypo-osmotic media in the presence of BafA1 (100 nM) or vehicle control (DMSO, 0.5% vol/vol) (6 h) (n = 5).

(E) Schematic of the cGAS-STING pathway and inhibitors used. A151 inhibits dsDNA sensing, Ru.251 and 4-sulfonic calix[6]arene are inhibitors of cGAS, and H151 inhibits STING. (F and G) qRT-PCR analysis of *Ifnb* (F) and *Cxcl10* (G) in WT and VRAC KO BMDMs incubated in hypo-osmotic media (50% $H_2O$ vol/vol in DMEM) for 6 h in the presence of H151 (10 µM), A151 (1 µM), or vehicle control (DMSO 0.5% vol/vol) (n = 4). (H and I) qRT-PCR analysis of *Ifnb* (H) and *Cxcl10* (I) in WT and VRAC KO BMDMs incubated in hypo-osmotic media for 6 h in the presence of Ru.251 (10 µM) or vehicle control (DMSO, 0.5% vol/vol) (n = 4). (J and K) qRT-PCR analysis of *Ifnb* (J) and *Cxcl10* (K) in WT and VRAC KO BMDMs incubated in hypo-osmotic media for 6 h in the presence of 4-sulfonic calix[6]arene (30 µM) or vehicle control (DMSO 0.5% vol/vol) (n = 4). (L) IFNβ release in the supernatant from WT and VRAC KO BMDMs stimulated with transfected poly dA:dT (1 µg ml⁻¹), mock-transfected, or treated with CMA (250 µg ml⁻¹) for 6 h (n = 5). (M) qRT-PCR analysis of *Sting1* in WT and VRAC KO BMDMs treated with siRNA targeting *Sting1*, or nontargeting, before incubation in iso-osmotic or hypo-osmotic media (6 h) (n = 3). (N) qRT-PCR analysis of *Cgas* in WT and VRAC KO BMDMs treated with siRNA targeting *Cgas*, or nontargeting, before incubation in iso-osmotic or hypo-osmotic media (6 h) (n = 3). (O and P) qRT-PCR analysis of *Ifnb* (O) and *Cxcl10* (P) in BMDMs from M and N (n = 3). *P < 0.05, **P < 0.01, ***P < 0.001, determined by a two-way ANOVA with Sidak's post hoc analysis (2B, 2D, 2M, and 2N) or Dunnett's post hoc analysis (2F–2K, 2O, and 2P). Values shown are the mean ± the SEM. Source data are available for this figure: SourceData F3.

(ds)DNA (Decout et al., 2021). Accumulation of cytosolic dsDNA is sensed by cGAS leading to the production of the STING agonist 2′3′-cGAMP and activation of STING (Fig. 3 E). To test involvement of cGAS-STING in the type I IFN response to cell volume changes, we used pharmacological inhibitors of the cGAS-STING pathway in WT and VRAC KO BMDMs in response to hypo-osmotic media (Fig. 3 E). Matching what we saw previously, VRAC KO BMDMs, which have lost control of cell volume regulation, showed a large increase in *Ifnb* and *Cxcl10* expression following incubation in hypo-osmotic media (Fig. 3, F and G). Pretreatment with the STING inhibitor H151 (Haag et al., 2018) or the immunosuppressive oligonucleotide A151, which antagonizes DNA sensors (Steinhagen et al., 2018), significantly blocked the induction of *Ifnb* and *Cxcl10* in response to hypo-osmotic media (Fig. 3, F and G), suggesting a DNA- and STING-dependent mechanism. However, the induction of a type I IFN response was found to occur independently of cGAS, as the cGAS inhibitors Ru.251 (Vincent et al., 2017) (Fig. 3, H and I) and 4-sulfonic calix[6]arene (Green et al., 2023) (Fig. 3, J and K) did not prevent cell swelling–induced *Ifnb* or *Cxcl10* expression. Further, antagonism of DNA-PK, another DNA sensor proposed to function upstream of STING (Taffoni et al., 2021), with NU7441 did not prevent hypo-osmotic media–induced *Ifnb* and *Cxcl10* expression (Fig. S3, A and B). WT and VRAC KO BMDMs released similar levels of IFNβ when transfected with synthetic dsDNA (poly dA:dT) or treated with the STING activator 10-carboxymethyl-9-acridanone (CMA) (Cavlar et al., 2013) (Fig. 3 L), suggesting loss of VRAC in conditions with normal cell volume regulation did not impact STING signaling. Treatment with H151, A151, or Ru.251, and to a lesser extent Nu7741, was also sufficient to block IFNβ release induced by transfection of poly dA:dT (Fig. S3 C). To further explore cGAS and STING, we then used RNAi in WT and VRAC KO BMDMs to knock down cGAS and STING. Converse to our findings with STING inhibitors, we found that despite >50% knockdown of cGAS and STING, respectively (Fig. 3, M and N), we did not observe any differences in hypo-osmotic induction of *Ifnb* or *Cxcl10* in WT or VRAC KO BMDMs (Fig. 3, O and P), suggesting either that H151 may have additional targets responsible or that partial STING inhibition is insufficient to block hypo-osmotic–induced type I IFN signaling. Thus, we conclude that STING may not alone be responsible for cell swelling induction of type I IFN signaling.

Our RNA-seq analysis revealed that suppression of mRNA processing pathways occurred following cell volume disruption and was further suppressed by VRAC KO (Fig. 1 F and Fig. S1 H).

Mutations in mRNA processing genes are associated with interferonopathies through RNA-sensing pathways (Crow and Stetson, 2022); thus, we also explored the potential that accumulation of unprocessed RNAs could be responsible for cell swelling–induced type I IFN signaling. To examine whether cell swelling impacted new protein translation, we performed a puromycin incorporation assay (Aviner, 2020). Puromycin is rapidly incorporated into newly synthesized proteins and can be detected using anti-puromycin antibodies, providing a measurement of protein translation. We added puromycin to WT and VRAC KO BMDMs after incubation in iso- or hypo-osmotic media for 3 h, or treatment with the protein translation inhibitor cycloheximide. Confirming our RNA-seq analysis, we found that new protein translation was reduced particularly in VRAC KO BMDMs incubated in hypo-osmotic media (Fig. 4 A), suggesting cell swelling had impacted mRNA translation pathways, but to a lesser extent than our positive control of cycloheximide (Fig. 4 A). We also observed stress granule formation in WT and VRAC KO BMDMs following incubation in hypo-osmotic media (Fig. 4 B), a common consequence of translation arrest (McCormick and Khaperskyy, 2017), detected by immunofluorescence of stress granule marker G3BP1. Therefore, to examine whether RNA sensing contributed to type I IFN signaling following cell volume disruption, we used RNAi targeting MAVS, a central downstream regulator of many RNA-sensing pathways (Chan and Gack, 2016). We found that despite a ∼50% knockdown of MAVS in WT and VRAC KO BMDMs (Fig. 4 C), induction of *Ifnb* and the ISG *Cxcl10* was unaltered (Fig. 4, D and E), suggesting that RNA sensing may not contribute to type I IFN signaling by cell volume changes. Together, these data suggest cell volume changes drive a DNA- and TBK1-dependent signaling pathway, independent of endosomal TLRs and RNA-sensing pathways, to produce an antiviral response which is exacerbated when RVD is blocked.

### Persistent cell swelling leads to delayed caspase-3–dependent cell death

We next tested whether cell swelling affected the viability of macrophages. While an MTT assay showed little impact on cell viability (Fig. 5 A), LDH release was significantly enhanced in VRAC KO macrophages incubated in hypo-osmotic media for 6 h (Fig. 5 B), suggesting persistent cell swelling eventually caused cell death. While not significant, WT BMDMs had a slight increase in LDH release after 6 h in hypo-osmotic media. We performed bright-field time-lapse imaging of WT and VRAC KO

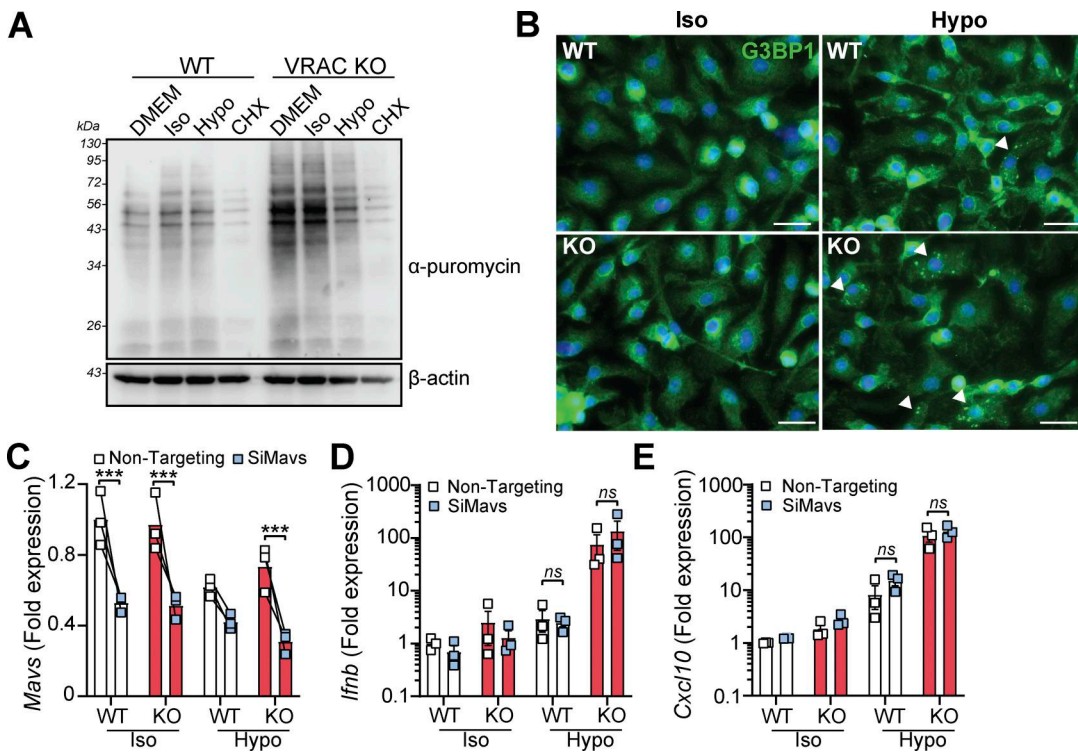

Figure 4. **Changes in cell volume induce translation arrest and stress granule formation, but type I IFN signaling is independent of RNA sensing through MAVS. (A)** Puromycin incorporation assay in WT and VRAC KO BMDMs incubated in DMEM, iso-osmotic media (50% PBS vol/vol in DMEM), hypo-osmotic media (50% H2O vol/vol/in DMEM), or CHXx (10 µg ml⁻¹) (4 h) (n = 3). **(B)** Immunofluorescence for the stress granule marker G3BP1 in WT and VRAC KO BMDMs incubated in iso-osmotic or hypo-osmotic media (2 h) (n = 4) (scale = 20 µm; arrowheads indicate stress granule–positive cells). **(C)** qRT-PCR analysis of *Mavs* in WT and VRAC KO BMDMs treated with siRNA targeting *Mavs*, or nontargeting, before incubation in iso-osmotic or hypo-osmotic media (6 h) (n = 3). **(D and E)** qRT-PCR analysis of *Ifnb* (D) and *Cxcl10* (E) in BMDMs from C (=3). ***P < 0.001, determined by a two-way ANOVA with Sidak's post hoc analysis (4C) or Tukey's post hoc analysis (4D and 4E). Values shown are the mean ± the SEM. CHX, cycloheximide. Source data are available for this figure: SourceData F4.

macrophages in hypo-osmotic media shown in Fig. S4. To further characterize the cellular pathways responsible for cell swelling–induced cell death, we pretreated WT and VRAC KO BMDMs with chemical inhibitors targeting various cell death pathways. Cell swelling–induced death was unaffected by inhibition of STING, RIPK1 (necroptosis), or inflammasome-associated caspase-1, but was significantly reduced by the pan-caspase inhibitor ZVAD (Fig. 5 C). Subsequently, we found that hypo-osmotic media induced activation of the apoptotic caspase-3, via a luminescence Caspase-Glo 3/7 assay (Fig. 5 D) and immunoblot for cleaved caspase-3 (Fig. 5 E). Notably, caspase-1 processing, a hallmark of inflammasome activation, was absent in response to this hypotonic shock (Fig. 5 E). Cell death and caspase-3 activation occurred independent of type I IFN signaling, as IFNAR KO BMDMs were not protected from hypo-osmotic–induced LDH release or caspase-3 cleavage (Fig. 5, F–I). These data show that cell swelling, despite not being severe enough to induce instant lysis, induces cell stress to initiate a delayed caspase-3–dependent cell death that is independent of type I IFN signaling.

## Cell volume disturbances potentiate STING signaling independently of cGAMP transfer

VRAC has previously been reported to modify antiviral responses by acting as a channel for 2′3′-cGAMP transport across plasma membranes, facilitating immune responses by propagating STING activation into bystander cells (Chen et al., 2021; Lahey et al., 2020; Zhou et al., 2020a). While we had found cGAS inhibitors and cGAS knockdown were unable to prevent the type I IFN response in hypo-osmotic conditions, because of the established role of VRAC in 2′3′-cGAMP transport, we explored the possibility that the increased type I IFN response that occurred in VRAC KO macrophages occurred due to defects in outward cGAMP transport, resulting in cGAMP accumulation (Fig. 6 A). VRACs are hexameric channels composed of a mixture of LRRC8 subunits, consisting of LRRC8A-E (Voss et al., 2014). While LRRC8A is essential for ion channel activity, and as a result RVD and volume regulation, LRRC8B-E are proposed to modify the selectivity of the pore for permeability to certain substrates (Ghouli et al., 2022). In the case for optimal cGAMP transport, VRAC channels must contain LRRC8A and LRRC8E (Chen et al., 2021; Lahey et al., 2020; Zhou et al., 2020a), and high expression of LRRC8D is proposed to inhibit 2′3′-cGAMP transport (Lahey et al., 2020). We therefore examined our RNA-seq dataset from our WT BMDMs (Fig. 1 A) to determine whether they expressed LRRC8 subunits compatible with cGAMP transport. BMDMs expressed high levels of LRRC8D, and less but equal amounts of LRRC8A, LRRC8B, and LRRC8C, and interestingly did not express any transcripts for LRRC8E (Fig. 6 B).

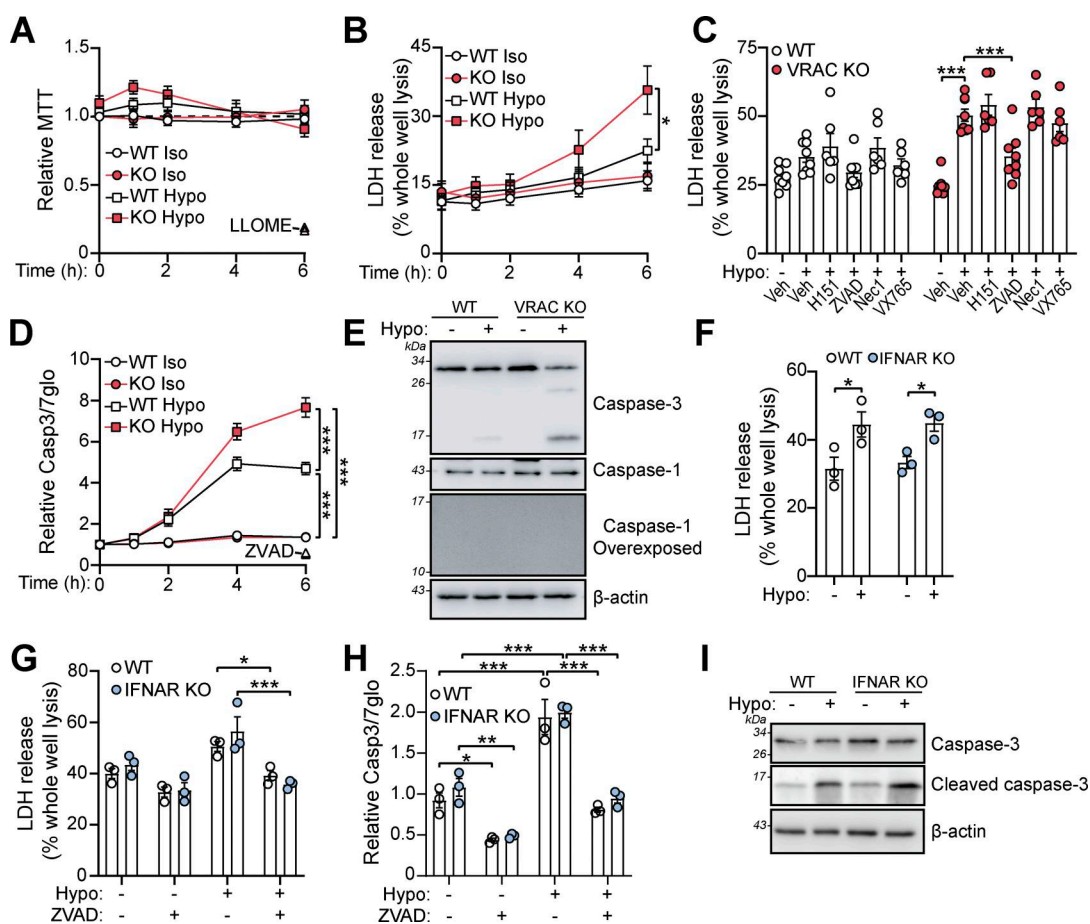

**Figure 5. Persistent cell swelling leads to caspase-3–dependent and type I IFN-independent cell death. (A and B)** WT and VRAC KO BMDMs were incubated in iso-osmotic or hypo-osmotic media for the indicated time points. Cell viability was assessed via an MTT assay (A) and LDH release into the supernatant (B). LLOME (1 mM, 6 h) was used as a positive control of cell death in A (n = 7). **(C and D)** WT and VRAC KO BMDMs were incubated in iso-osmotic or hypo-osmotic media in the presence of STING inhibitors (H151, 10 µM), pan-caspase inhibitors (ZVAD-FMK, 50 µM), necroptosis inhibitors (RIPK1 inhibitor necrostatin, nec1, 50 µM), caspase-1/11 inhibitors (VX765, 10 µM), or vehicle control (DMSO, 0.5% vol/vol) (n = 6–8). (D) Caspase-Glo 3/7 assay (on combined cells and supernatant) in WT and VRAC KO BMDMs incubated in iso-osmotic or hypo-osmotic media for the indicated time points, or in the presence of ZVAD (50 µM, 6 h) (n = 4). **(E)** Western blot for caspase-3 and caspase-1 in WT and VRAC KO BMDMs incubated in iso- or hypo-osmotic media (6 h) (n = 3). **(F and G)** LDH release from WT and IFNAR KO BMDMs incubated in iso- or hypo-osmotic media (6 h) (F) (n = 3), and in the presence of ZVAD (50 µM) or vehicle control (DMSO, 0.5% vol/vol) (G) (n = 3). **(H)** Caspase-Glo 3/7 assay (on combined cells and supernatant) in WT and IFNAR KO BMDMs incubated in iso-osmotic or hypo-osmotic media in the presence of ZVAD (50 µM) or vehicle control (DMSO, 0.5% vol/vol) (6 h) (n = 3). **(I)** Western blot for caspase-3 in WT and IFNAR KO BMDMs incubated in iso- or hypo-osmotic media (6 h) (n = 3). *P < 0.05, **P < 0.01, ***P < 0.001, determined by a two-way ANOVA with Sidak's post hoc analysis (5C and 5E), uncorrected Fisher's LSD (5D), or Tukey's post hoc analysis (5G and 5H). Values shown are the mean ± the SEM. Source data are available for this figure: SourceData F5.

Therefore, the low expression of LRRC8E and high expression of LRRC8D suggest that VRAC in our BMDMs would be a poor conductor of cGAMP. To test this, we incubated WT and VRAC KO BMDMs in isotonic and hypotonic conditions for 3 h in the presence of extracellular 2'3'-cGAMP and examined induction of *Ifnb* and *Cxcl10*. Cells were incubated for only 3 h to limit the contribution of hypo-osmotic media alone to the *Ifnb* and *Cxcl10* response. In parallel, as a positive control, we performed the same experiment in a mouse fibroblast cell line L929, which has been shown to express LRRC8E and conduct cGAMP efficiently (Zhou et al., 2020a). L929 cells upregulated both *Ifnb* and *Cxcl10* in response to extracellular cGAMP, and this was significantly enhanced by incubation in hypo-osmotic media (Fig. 6, C and D). However, WT and VRAC KO BMDMs exhibited only weak responses to extracellular cGAMP, which was not enhanced by

incubation in hypo-osmotic media (Fig. 6, C and D). We then examined whether cell volume influenced STING signaling, using a small molecule activator of murine STING CMA. CMA directly binds to STING, independent of cGAS and 2'3'-cGAMP (Cavlar et al., 2013), also avoiding any conflating involvement of 2'3'-cGAMP transport between cells through VRAC. We found that hypo-osmotic media potentiated CMA-induced STING signaling, with enhanced *Ifnb* transcription (Fig. 6 E) and phosphorylation of downstream TBK1 and IRF3 (Fig. 6, F–H). In both cases, VRAC KO BMDMs exhibited a further enhancement (Fig. 6, E–H). Co-treatment with brefeldin A, which prevents STING translocation from the ER to the ERGIC (Ogawa et al., 2018), completely removed the response, showing that STING potentiation was not through cell swelling causing mislocalization of STING to the ERGIC (Fig. 6, F–H). Further, STING dimerization, a

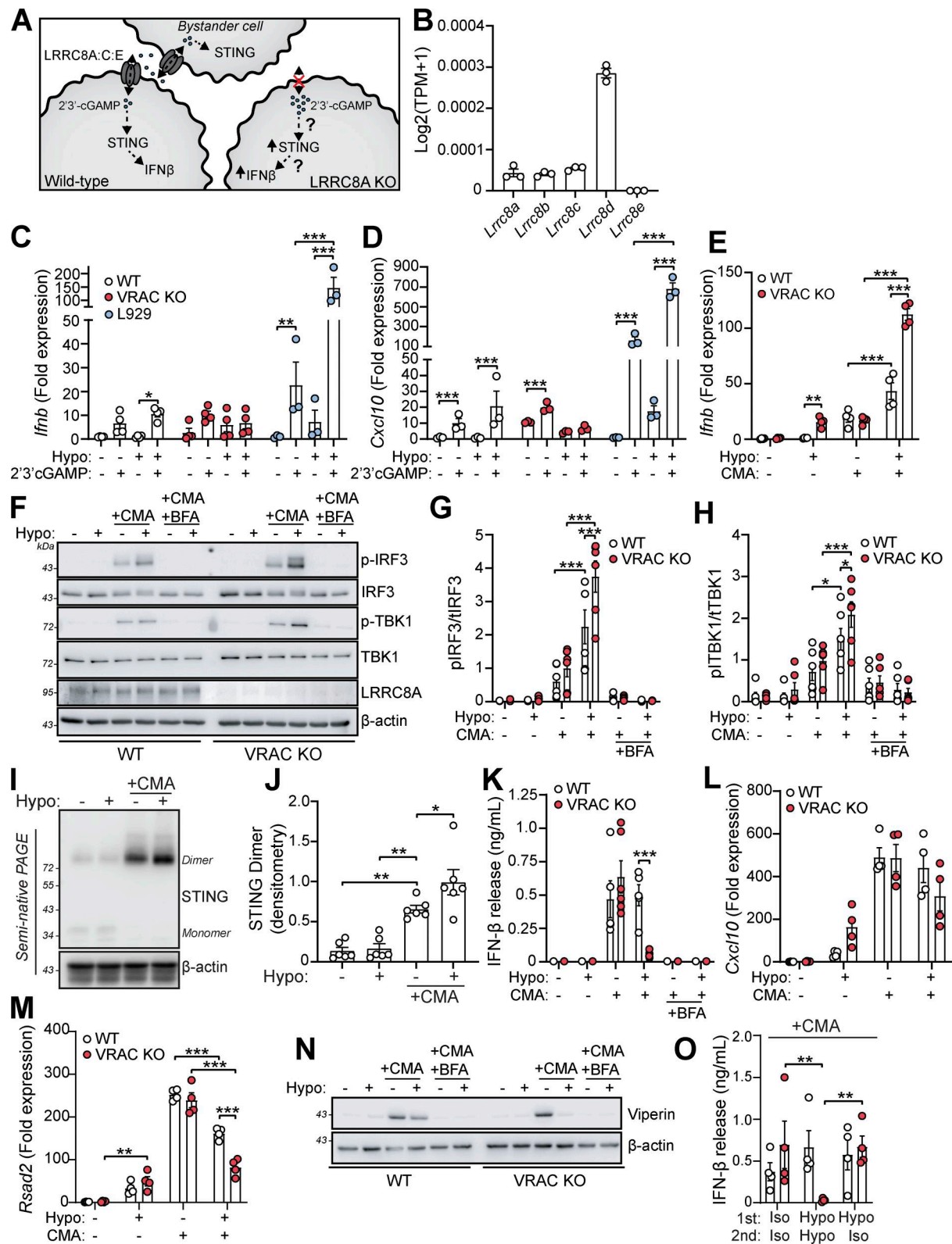

Figure 6. **Cell volume disturbances and loss of VRAC potentiate STING signaling independent of cGAMP transport. (A)** Schematic of the proposed role of VRAC in cGAMP transport. VRAC containing LRRC8A, LRRC8C, and LRRC8E can act as a conduit for 2'3'-cGAMP allowing transport between extracellular space. **(B)** Number of mRNA transcripts of LRRC8 subunit genes in WT BMDMs from RNA-seq in Fig. 1 A (TPM = transcript per million). **(C and D)** qRT-PCR analysis of *Ifnb* (C) and *Cxcl10* (D) in WT and VRAC KO BMDMs, and L929 mouse fibroblasts, following incubation in isotonic or hypotonic media with or without extracellular cGAMP (5 µg ml$^{-1}$) for 3 h ($n$ = 4 for BMDMs, $n$ = 3 for L929 s). **(E)** qRT-PCR analysis for *Ifnb* of WT and VRAC KO BMDMs incubated in iso- or hypo-osmotic media in the presence of the direct murine STING agonist CMA (250 µg ml$^{-1}$) or vehicle control (1% DMSO vol/vol) (6 h) ($n$ = 4). **(F)** Western blot (F) of phosphorylated (p-) and total IRF3 and TBK1 in WT and VRAC KO BMDMs incubated in iso-osmotic or hypo-osmotic media in the presence of CMA (250 µg ml$^{-1}$)

and or BFA (10 µg ml⁻¹) or vehicle control (DMSO, 0.2% vol/vol) (3 h) (n = 6). **(G and H)** Densitometry of pIRF (G) and pTBK1 (H) from F (n = 6). **(I)** STING dimerization assessed by semi-native PAGE in WT BMDMs incubated in iso-osmotic or hypo-osmotic media in the presence of CMA (250 µg ml⁻¹) or vehicle control (DMSO, 1% vol/vol) (4 h) (n = 6). **(J)** Densitometry of STING dimers in I (n = 6). **(K)** IFNβ release in the supernatant of WT and VRAC KO BMDMs incubated in iso-osmotic or hypo-osmotic media in the presence of CMA (250 µg ml⁻¹) or vehicle control (DMSO, 1% vol/vol) and or BFA (10 µg ml⁻¹) (6 h) (n = 6). **(L and M)** qRT-PCR analysis for Cxcl10 (L) and Rsad2 (M) from experiment in E. **(N)** Western blot of viperin in WT and VRAC KO BMDMs incubated in BMDMs in iso-osmotic or hypo-osmotic media in the presence of CMA (250 µg ml⁻¹) and or BFA (10 µg ml⁻¹) or vehicle control (DMSO, 0.2% vol/vol) (6 h) (n = 6). **(O)** WT and VRAC KO BMDMs were incubated in iso- or hypo-osmotic media + CMA (250 µg ml⁻¹) for 3 h (first), before the media were replaced with fresh iso- or hypo-osmotic media (second) without CMA for a further 3 h. IFNβ release shown is from the second media incubation (n = 4). *P < 0.05, **P < 0.01, ***P < 0.001, determined by a one-way ANOVA with Dunnett's post hoc analysis (6J) or two-way ANOVA with Dunnett's post hoc analysis (6C and 6D), Sidak's post hoc analysis (6E, 6 L, and 6M), or Tukey's post hoc analysis (6F–6 G, 6K, and 6O). Values shown are the mean ± the SEM. BFA, brefeldin A. Source data are available for this figure: SourceData F6.

key proximal event following STING activation, was enhanced by incubation in hypo-osmotic media (Fig. 6, I and J), demonstrating that cell swelling was acting to modify STING responses upstream of STING. However, despite the increase in STING signaling, IFNβ release (Fig. 6 K) and the induction and expression of downstream ISGs Cxcl10 and Rsad2 were not enhanced and were actually significantly reduced (Fig. 6, L–N). We propose this reduction occurs due to the reduction in new protein translation observed in BMDMs following cell swelling (Fig. 4 A and Fig. S1 H). Supporting this, replacement of the media with DMEM after stimulation with CMA in hypo-osmotic media fully restored IFNβ release (Fig. 6 O). Therefore, these data suggest that cell volume control has a significant role in directing STING signaling, which occurs independently of VRAC-mediated cGAMP transfer.

## Cell volume regulation modifies the inflammatory response to pathogen-associated signals

Since we had established changes in cell volume promoted an inflammatory phenotype in macrophages and enhanced CMA-induced STING signaling, we considered whether cell volume changes might synergize with other inflammatory triggers, such as PAMPs, to modify the inflammatory response further. This is particularly relevant as infection modifies the tissue microenvironment with signals known to modify cell volume such as edema, hypoxia, oxidative stress, and metabolic dysfunction (Chen et al., 2019; Eltzschig and Carmeliet, 2011; Friard et al., 2021). We therefore stimulated WT and VRAC KO BMDMs with hypo-osmotic media in the presence of the TLR agonists: poly I:C (TLR3), LPS (TLR4), imiquimod (TLR7), or CpG-DNA (TLR9), which drive the induction of pro-inflammatory cytokines through activation of the transcription factors NF-κB and IRFs (Fitzgerald and Kagan, 2020). In the supernatant of stimulated BMDMs, we measured release of IL-6, a cytokine dependent on NF-κB signaling, and IFNβ, a cytokine dependent on IRF signaling to see whether response to the signaling pathways had been modified. Interestingly, we found that across all PAMPs tested, IFNβ release was significantly increased by inducing changes in cell volume (Fig. 7, A and D). In response to poly I:C or LPS, while there was no difference in IFNβ release between WT and VRAC KO macrophages in isotonic conditions, stimulation in hypo-osmotic media enhanced IFNβ release in both WT and VRAC KO BMDMs, which was greater in VRAC KO cells (Fig. 7, A and B). Strikingly, both imiquimod and CpG-DNA only induced IFNβ release in VRAC KO BMDMs under hypotonic conditions

(Fig. 7, C and D), suggesting loss of cell volume regulation completely altered the TLR7 and TLR9 responses. No change in IL-6 release was observed following PAMP stimulation in hypotonic conditions, except for poly I:C, which exhibited the same pattern as the poly I:C–induced IFNβ response (Fig. 7, E and H). Because endosomal TLR7 and TLR9 showed large changes in VRAC KO BMDMs in hypo-osmotic media, we examined whether lysosomal damage or dysfunction was occurring during cell swelling. However, punctate galectin-3 staining, a marker of lysosomal damage, was absent in WT and VRAC KO BMDMs incubated in hypo-osmotic media (Fig. S5 A). Further, lysosomal trafficking and function were assessed using DQ-BSA uptake, which showed no changes between WT and VRAC KO BMDMs in either iso- or hypo-osmotic media (Fig. S5 B), suggesting normal endolysosomal function in these conditions. Of note, while hypo-osmotic media caused partial translational arrest that limited IFNβ release and ISG production in response to CMA (Fig. 6, K and L), this was absent following treatment with PAMPs, demonstrated by high levels of cytokine release, suggesting PAMPs could override translational arrest induced here. Further, PAMPs were able to abrogate hypo-osmotic cell death, especially TLR7 and TLR9 stimulants, which completely removed hypo-osmotic LDH release (Fig. 7 I). Together, these data suggest that cell volume regulation controls the inflammatory response to various PAMPs, with cell volume increases driving enhanced type I IFN responses.

Since hypotonicity-induced cell swelling and RVD modified the macrophage response to PAMPs, we investigated whether inducing a reduction in cell volume with hypertonic solutions also altered the inflammatory response. To achieve this, we incubated WT BMDMs in either DMEM, hypo-osmotic DMEM (50% vol/vol H₂O), isotonic DMEM (50% vol/vol 150 mM NaCl), or hyperosmotic DMEM (50% vol/vol 300 mM NaCl), stimulated with poly I:C, LPS, imiquimod, or CpG-DNA, and measured IL-6 and IFNβ release. We found that in contrast to hypotonicity, hypertonicity was effective at suppressing IL-6 release in response to poly I:C, LPS, imiquimod, and CpG-DNA (Fig. 7, J–M). However, IFNβ release was only altered with poly I:C treatment (Fig. 7 N), with LPS-induced IFNβ unchanged (Fig. 7 O) and no IFNβ detected following imiquimod or CpG-DNA treatment (data not shown). Together, these findings suggest that cell volume acts as an additional layer of danger sensing in macrophages that shapes and tunes the nature of immune responses to pathogens, with cell swelling promoting IFNβ signaling and cell shrinkage suppressing IL-6 responses.

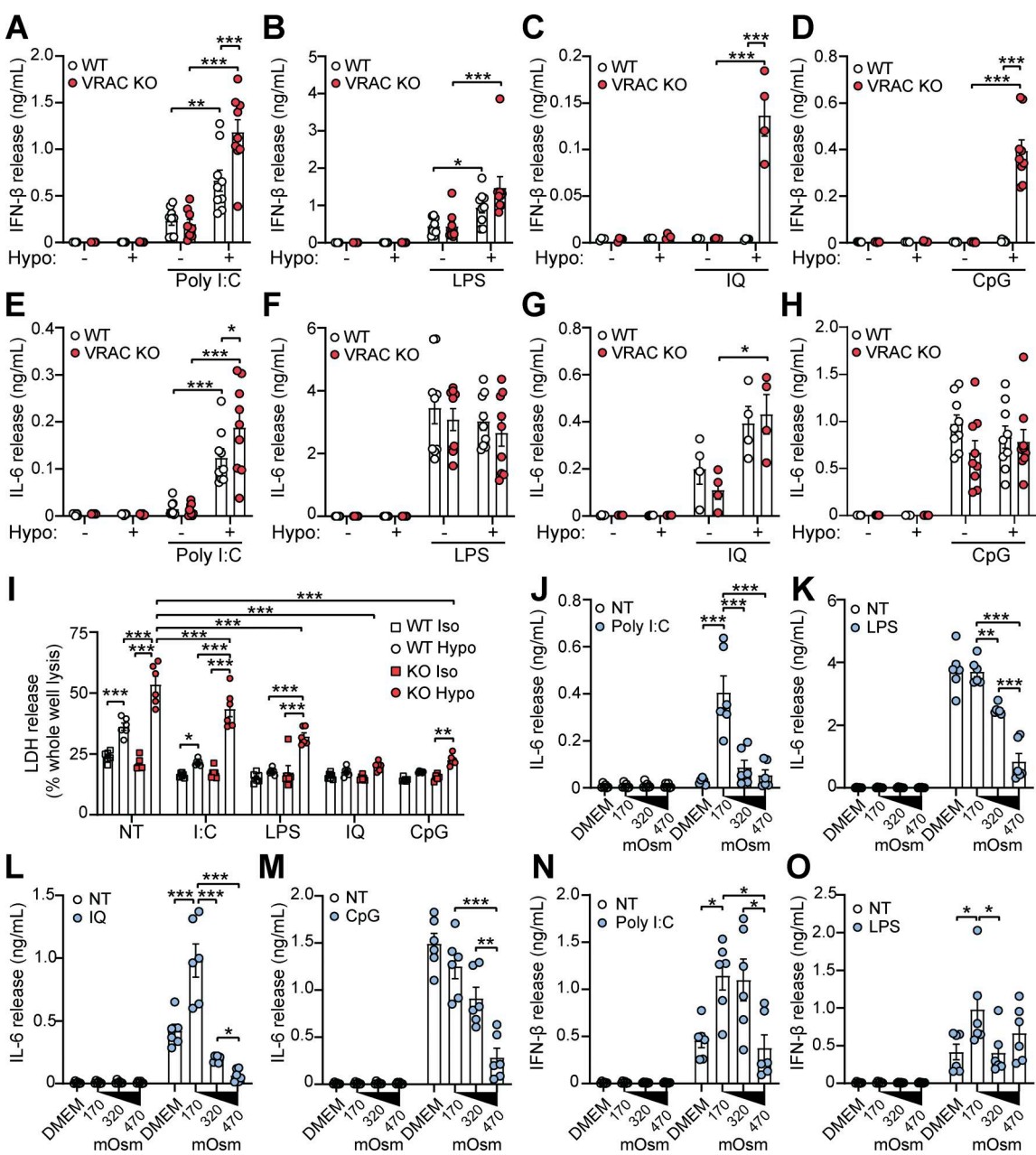

Figure 7. **Cell volume regulates type I interferon production in response to diverse pathogen-mediated molecular patterns. (A–D)** IFNβ release in the supernatant of WT and VRAC KO BMDMs incubated in DMEM or hypo-osmotic media (50% vol/vol H$_2$O in DMEM) and treated with poly I:C (5 μg ml$^{-1}$) (A), LPS (1 μg ml$^{-1}$) (B), imiquimod (IQ, 25 μM) (C), or CpG-DNA (CpG, 5 μg ml$^{-1}$) (D) for 6 h ($n$ = 9 for poly I:C, LPS, CpG; $n$ = 4 for IQ). **(E–H)** IL-6 release in the supernatant of WT and VRAC KO BMDMs incubated in DMEM or hypo-osmotic media (50% vol/vol H$_2$O in DMEM) and treated with poly I:C (5 μg ml$^{-1}$) (E), LPS (1 μg ml$^{-1}$) (F), imiquimod (25 μM) (G), or CpG-DNA (5 μg ml$^{-1}$) (H) for 6 h ($n$ = 9 for poly I:C, LPS, CpG; $n$ = 4 for IQ). **(I)** LDH release from WT and VRAC KO BMDMs incubated in iso-osmotic or hypo-osmotic media in the presence of poly I:C (I:C, 5 μg ml$^{-1}$), LPS (1 μg ml$^{-1}$), IQ (IQ, 25 μM), or CpG-DNA (CpG, 5 μg ml$^{-1}$) for 6 h ($n$ = 6). **(J–M)** IL-6 release in the supernatant of WT BMDMs incubated in either DMEM, hypo-osmotic media (50% vol/vol H$_2$O in DMEM), isotonic media (50% vol/vol 150 mM NaCl), or hyperosmotic media (50% vol/vol 300 mM NaCl) and treated with poly I:C (5 μg ml$^{-1}$) (J), LPS (1 μg ml$^{-1}$) (K), imiquimod (IQ, 25 μM) (L), or CpG-DNA (CpG, 5 μg ml$^{-1}$) (M) for 6 h ($n$ = 6). **(N and O)** IFNβ release in the supernatant of WT BMDMs incubated in either DMEM, hypo-osmotic media (50% vol/vol H$_2$O in DMEM), isotonic media (50% vol/vol 150 mM NaCl), or hyperosmotic media (50% vol/vol 300 mM NaCl) and treated with poly I:C (5 μg ml$^{-1}$) (N) or LPS (1 μg ml$^{-1}$) (O) for 6 h ($n$ = 6). *P < 0.05, **P < 0.01, ***P < 0.001, determined by a two-way ANOVA with Sidak's post hoc analysis (7A–7H) or Tukey's post hoc analysis (7I), or by a one-way ANOVA with Tukey's post hoc analysis (7J–7O). Values shown are the mean ± the SEM.

## Cell volume control impacts antiviral responses to influenza A virus

Since we found that cell swelling induced and influenced type I IFN signaling and induction of antiviral genes, we then examined whether VRAC and cell volume control were important for

antiviral responses to pathogenic virus infection. To test this, we infected WT and VRAC KO BMDMs with the RNA virus influenza A virus (IAV, strain H1N1; A/PR/8/34). Since it is an RNA virus, cGAS is not involved (Holm et al., 2016) limiting potential 2′3′-cGAMP transport through VRAC channels. Matching our findings

Figure 8. **Macrophage cell volume control influences antiviral responses to influenza A (IAV) infection. (A–C)** WT and VRAC KO BMDMs were infected with IAV (MOI 10, 24 h) or mock-treated. qRT-PCR analysis for *Ifnb* (A), *Cxcl10* (B), and *Rsad2* (C) (*n* = 5). **(D)** CXCL10 ELISA from supernatant from IAV (MOI 10, 24 h) or mock-infected WT and VRAC KO BMDMs (*n* = 6). **(E and F)** qRT-PCR analysis of the IAV-derived transcripts M1 (E) and M2 (F) from experiments (A–C) (*n* = 5). **(G)** qRT-PCR analysis for *Lrrc8a* in WT BMDMs infected with IAV (MOI 5, 6 h) (*n* = 5). **(H)** Western blot for viperin in WT BMDMs treated with empty Lipofectamine 3000 (Lipo, 0.5% vol/vol) complexes in the presence of the membrane-stabilizing agent punicalagin (Puni, 50 μM) or vehicle control (DMSO, 0.5% vol/vol) (6 h) (*n* = 3). *P < 0.05, **P < 0.01, ***P < 0.001, determined by a two-way ANOVA with Sidak's post hoc analysis (8A–8F), or by Student's *t* test (8 G). Values shown are the mean ± the SEM. Source data are available for this figure: SourceData F8.

with hypotonic cell swelling, VRAC KO BMDMs exhibited enhanced antiviral responses to IAV infection, exhibiting enhanced production of *Ifnb* and the ISGs *Cxcl10* and *Rsad2* (Fig. 8, A–C). CXCL10 release in the supernatant was also significantly enhanced in VRAC KO macrophages (Fig. 8 D). We then assessed whether viral burden was altered between WT and VRAC KO macrophages by examining expression of the viral transcripts that encode IAV proteins M1 and M2. M1 and M2 transcripts were identified exclusively in IAV-infected macrophages, which was not significantly different between WT and VRAC KO BMDMs (Fig. 8, E and F). Interestingly, we also observed a significant downregulation of LRRC8A transcripts in WT BMDMs infected with IAV (Fig. 8 G), suggesting that IAV could also be downregulating cell volume control mechanisms. Viral fusion with the plasma membrane has been proposed to promote type I IFN signaling independent of viral components (Holm et al., 2012); thus, we propose that changes in cell volume control could influence plasma membrane dynamics to influence type I IFN signaling. To test whether membrane perturbations influence type I IFN signaling, we treated BMDMs with empty Lipofectamine complexes to disrupt the membrane, in combination with the membrane-stabilizing agent punicalagin (Martin-Sanchez et al., 2016). We found that Lipofectamine (5% vol/vol) alone was sufficient to produce high levels of viperin, indicating type I IFN signaling was occurring, which was completely blocked upon membrane stabilization with punicalagin (Fig. 8 H). These data suggest that VRAC and cell volume control are critical mediators in initiating antiviral defense to influenza A.

### Cell volume regulates inflammation *in vivo*

We next examined whether regulation of macrophage cell volume played a role in shaping inflammatory responses *in vivo*. To

do this, we used an established mouse model of macrophage activation syndrome (MAS), where a systemic inflammatory response is induced by five intraperitoneal (i.p.) injections of CpG-DNA over 10 days (Behrens et al., 2011). This model recapitulates several features of hyperinflammatory disease, including cytokine storm and multi-organ dysfunction, with inflammation occurring throughout the body (Behrens et al., 2011; Gleeson et al., 2024). We rationalized that the induction of highly inflammatory tissue states over 10 days would create established inflammatory environments where cell volume would be impacted. We therefore used the model CpG-induced MAS with our conditional VRAC KO mice (CX3CR1-cre[+/−]; LRRC8A[flox/flox], referred to here as VRAC[Cx3cr1-KO]) and littermate WT mice (CX3CR1-cre[−/−]; LRRC8A[flox/flox]), treated as shown in Fig. 9 A. Matching previous reports, repeated administration of CpG-DNA induced transient weight loss in both WT and VRAC[Cx3cr1-KO] mice, although this was only significant in VRAC[Cx3cr1-KO] mice (Fig. 9, B and C). CpG-DNA treatment also induced splenomegaly (Fig. 9 D), hepatomegaly (Fig. 9 E), and hyperferritinemia (Fig. 9 F), which was the same between WT and VRAC[Cx3cr1-KO] mice. These data suggest that loss of macrophage cell volume control did not affect the development of MAS. We then examined whether the nature of the cytokine storm was altered between WT and VRAC[Cx3cr1-KO] mice. Repeated administration of CpG-DNA caused the induction of multiple plasma cytokines, including IFNγ (Fig. 9 G), IL-18 (Fig. 9 H), TNF (Fig. 9 I), IL-6 (Fig. 9 J), and IL-10 (Fig. 9 K). Notably in VRAC[Cx3cr1-KO] mice, CpG treatment significantly elevated plasma IL-18, which was doubled compared with WT littermates (Fig. 9 H). While type I IFNs were not detected in the plasma of CpG-treated mice (data not shown), IL-18 is known to be induced by interferon

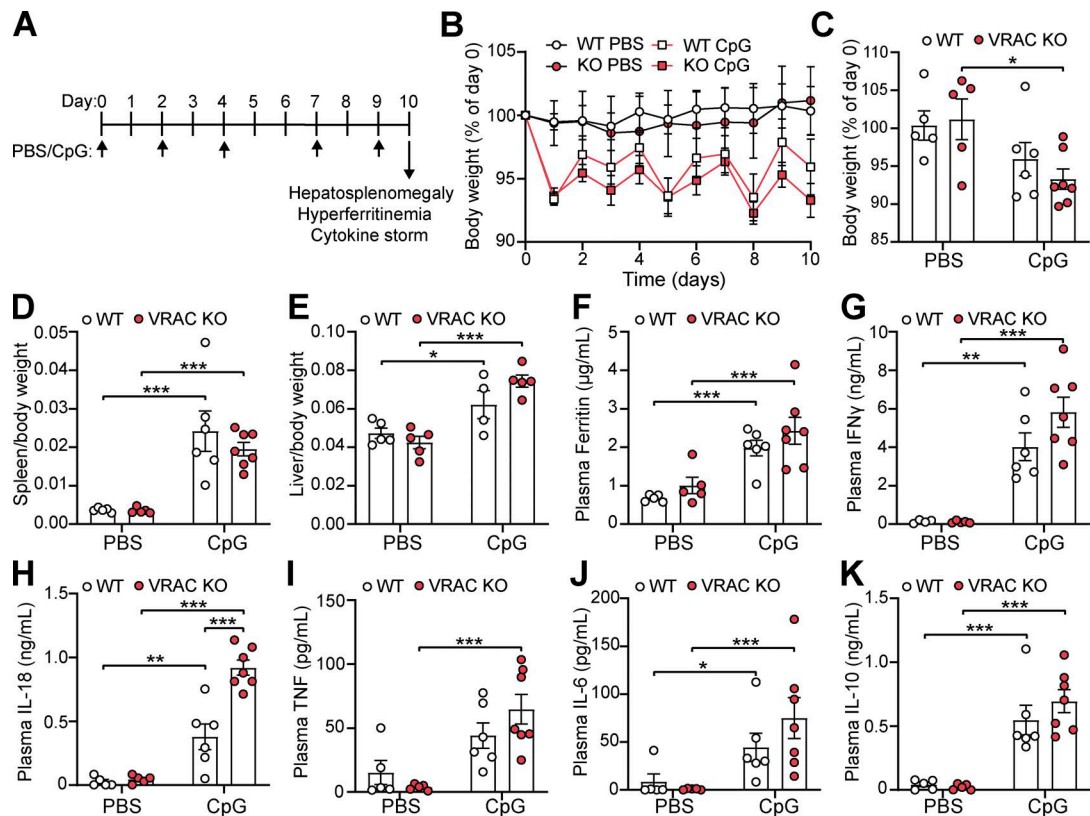

**Figure 9. Cell volume control mechanisms regulate inflammation in a murine model of hyperinflammation. (A)** Schematic of the murine model of CpG-induced hyperinflammation. CpG-DNA (2 mg kg$^{-1}$), or PBS, was administered by i.p. injection on days 0, 2, 4, 7, and 9. On day 10, tissues were taken and hyperinflammatory disease was assessed by hepatosplenomegaly, hyperferritinemia, and cytokine storm. **(B)** Body weight of WT and VRAC$^{Cx3cr1-KO}$ mice (VRAC KO) treated with PBS or CpG-DNA expressed as a percentage of day 0 ($n$ = 5–7). **(C)** Body weight at day 10 from B ($n$ = 5–7). **(D)** Splenic weight normalized to body weight ($n$ = 5–7). **(E)** Liver weight normalized to body weight ($n$ = 4–5). **(F)** Plasma levels of ferritin ($n$ = 5–7). **(G–K)** Plasma concentration of IFNγ (G), IL-18 (H), TNF (I), IL-6 (J), and IL-10 (K) ($n$ = 5–7). *$P < 0.05$, **$P < 0.01$, ***$P < 0.001$, determined by a two-way ANOVA with Sidak's post hoc analysis. Values shown are the mean ± the SEM.

signaling, thus suggesting that a loss of cell volume regulation in VRAC$^{Cx3cr1-KO}$ mice contributes to dysregulated inflammatory responses. Therefore, while VRAC$^{Cx3cr1-KO}$ mice did not exhibit any significant worsening of disease phenotypes, we propose that control of macrophage cell volume does play a role in coordinating inflammatory responses as loss of macrophage cell volume control resulted in enhanced systemic cytokine production.

## Discussion

Tiered inflammatory responses related to inflammasome activation in response to increasing DAMP-mediated stress are proposed previously for mechanisms of IL-1β release (Lopez-Castejon and Brough, 2011), and more recently for DNA sensing (Emming and Schroder, 2019). Related to this last example, the evidence presented in this manuscript and in the literature suggests that a tiered inflammatory response to cell volume changes in macrophages may also occur. In response to extreme hypotonicity and cell swelling, macrophages respond with activation of the NLRP3 inflammasome via a process that is dependent on the RVD and the Cl$^-$ channel VRAC (Compan et al., 2012; Green et al., 2020; Wu et al., 2023). Here, we demonstrate an additional danger-sensing pathway in macrophages to milder

hypotonic stresses that initiate type I IFN expression and signaling that is enhanced following loss of cell volume control. Studying hypotonicity, we were also able to establish that changes in cell volume synergize with other PAMP stimuli to modify and shape inflammatory responses—highlighting cell volume regulation as a critical aspect of inflammatory signaling. Understanding disruptions in the tissue microenvironment leading to alterations in cell volume is therefore an important consideration in our understanding of inflammation and disease pathogenesis.

In addition to our studies in macrophages establishing the role of VRAC in RVD-dependent NLRP3 activation (Green et al., 2020), VRAC-dependent RVD is also reported to be important in T cells where loss of RVD impairs TCR signaling, resulting in impaired T cell activation, cytokine production, proliferation, and impaired antiviral immunity to acute lymphocytic choriomeningitis virus Armstrong infection (Wang et al., 2023). In addition to controlling changes in cell volume, VRAC is demonstrated to have additional important roles in antiviral immunity through acting as a plasma membrane channel for transport of 2′3′-cGAMP (Chen et al., 2021; Lahey et al., 2020; Zhou et al., 2020a). Global KO of LRRC8E, the subunit important for 2′3′-cGAMP transport, inhibited induction of antiviral genes and viral clearance in mice infected with the herpes simplex

virus (HSV)-1 (Zhou et al., 2020a). KO of LRRC8C in T cells also prevented 2′3′-cGAMP transport but enhanced antiviral immunity to influenza infection (Concepcion et al., 2022). Furthermore, it has recently been demonstrated that several viruses, including HSV-1, VACV, and ZIKV, induce degradation of LRRC8A to prevent cGAMP transport and restrain antiviral defense (Blest et al., 2024). Here, we also show that *Lrrc8a* transcripts are also reduced following IAV infection in BMDMs. The impact of virus-induced VRAC depletion on cell volume control, however, remains untested. Until now, no studies have reported a role of LRRC8A or VRAC in macrophages regulating RVD that contributes to IFN signaling. Here, we report that VRAC influenced inflammatory and antiviral signaling through cell volume control in macrophages, independent of transfer of 2′3′-cGAMP between cells, but dependent upon DNA sensing and TBK1/IKKε signaling. Furthermore, we show that loss of VRAC influences antiviral responses to the RNA virus IAV, where cGAS is not directly involved in pathogen sensing (Holm et al., 2016), positioning cell volume control itself as important regulator of antiviral inflammation. Since viral infection reduces VRAC expression (Blest et al., 2024), and consequently potentially cell volume control, it is possible that the immune system has evolved to sense a loss of cell volume control as a marker of viral infection, explaining the pro-inflammatory and antiviral response observed in this study. While pharmacological inhibition of DNA sensing, TBK1/IKKε, and STING was effective at blocking cell swelling–induced type I IFN signaling, we found that RNAi-mediated knockdown of STING, or the central regulator of RNA signaling MAVS, had no effect. While there is potential involvement of STING, our data show a clear lack of cGAS involvement, suggesting cell volume disruption could be activating a cGAS-independent STING pathway similar to those observed previously (Holm et al., 2016; Suschak et al., 2016; Unterholzner and Dunphy, 2019). Our findings suggest potential redundancy between nucleic acid–sensing pathways in coordinating type I IFN responses following cell volume disruption, similar to that seen with viral sensing (Cai et al., 2021). A caveat of our study is that we used a pharmacological and RNAi approach to identify responsible pathways. Future studies using full genetic KOs (such as of STING and TBK1) could help clarify and reveal the full nature of nucleic acid sensing occurring following cell volume disruption.

Mechano-immunity has been coined to describe the interplay between cellular mechanotransduction and inflammation and is becoming increasingly recognized as an important factor in inflammatory disease (Bezbradica and Bryant, 2024). Changes in the biophysical properties of the tissue environment in inflammatory conditions are known to alter the mechanical cues sensed by immune cells, resulting in altered cellular responses including cell activation, cytokine production, and proliferation among others (Du et al., 2023). Pro-inflammatory signaling is known to be initiated in macrophages by mechanosensitive ion channels, such as Piezo1 and TRPV4 (Atcha et al., 2021; Lee et al., 2022), and can synergize with PAMP/DAMP-mediated inflammation to sensitize and augment inflammatory responses (Geng et al., 2021; Ran et al., 2023). Here, we identified that changes in macrophage cell volume, particularly following loss of cell

volume control, led to the induction of intracellular inflammatory signaling, including the NF-κB and type I IFN pathways, and augmented TLR-mediated inflammatory responses. Thus, in addition to Piezo1 and TRPV4, VRAC can also be considered a key mediator of mechano-immunity in macrophages, coordinating cell volume changes to translate disturbances in the inflammatory environment to tune the immune response. It is possible that the biomechanical changes that occur at the plasma membrane during the process of cell swelling and regulated volume decrease could activate additional mechanosensory pathways. Indeed, plasma membrane perturbations induced by osmotic stress can activate both Piezo1 (Syeda et al., 2016) and TRPV4 (Toft-Bertelsen and MacAulay, 2021). Interestingly, plasma membrane disruption from fusion of the viral capsid is sufficient to drive a $Ca^{2+-}$ and STING-dependent type I IFN response (Holm et al., 2012; Noyce et al., 2011), suggesting that cell volume changes may function through similar conserved pathways. Supporting this, our study has found that plasma membrane disruption with Lipofectamine promoted ISG expression in macrophages, which was blocked following membrane stabilization using punicalagin (Martin-Sanchez et al., 2016). Moreover, membrane integrity changes following HSV-1 infection were recently demonstrated to restrain the cGAS-STING pathway by inducing cGAMP degradation via the sphingomyelinase SMPDL3B (Wang et al., 2025), highlighting that pathogens could have also evolved to evade and regulate antiviral host responses promoted by membrane disturbances. Future studies will further elucidate the interplay between VRAC and changes in cell volume and plasma membrane integrity in the mechano-immunity response of macrophages.

In our experiments, we observe that the VRAC-dependent RVD is important for NLRP3 activation in response to severe hypo-osmotic stress (Green et al., 2020), and that in response to milder hypo-osmotic stress, VRAC-dependent RVD appears to restrain inflammatory responses, highlighting a complex regulation of inflammation by cell volume change. While the osmolarities tested *in vitro* in these studies are perhaps more severe, our goal was to identify fundamental concepts of how cell volume changes are sensed by macrophages to influence inflammation. Cell volume disruption is induced in a range of complex pathophysiological conditions in addition to osmolarity stress, such as edema, oxidative stress, hypoxia/ischemia, hypothermia, and metabolic dysfunction (Chen et al., 2019; Eltzschig and Carmeliet, 2011; Friard et al., 2021; Hoffmann et al., 2009; Osei-Owusu et al., 2018), highlighting a need to understand how cell volume modulation in these conditions can impact inflammatory responses. Our *in vivo* model of repeated CpG-DNA–induced cytokine storm syndrome and hyperinflammation, similar to those following severe viral infection such as IAV and SARS-CoV-2 (Fajgenbaum and June, 2020), with our VRAC[CX3CR1-KO] mice demonstrates that cell volume control does impact inflammatory and immune responses in highly inflammatory tissue environments. However, while we observe changes in cytokine storm development, one limitation is that our study does not directly show that changes in macrophage activation are the cause of the altered cytokine storm and instead could be the result of indirect mechanisms. Specifically, plasma IL-18

levels were significantly elevated in VRAC[CX3CR1-KO] mice, an IFN-inducible and inflammasome-dependent cytokine (Landy et al., 2024). Why IL-18 is particularly upregulated is an interesting future avenue to explore, potentially through increased inflammasome activation, interferon signaling, or altered membrane permeability to cell volume changes, and highlights the complexity of studying cell volume responses with *in vivo* diseases. Future studies using cell-restricted VRAC KO mice will help reveal roles of cell volume control in disease. Therapeutically, hyperosmotic solutions have been utilized to treat several conditions, predominantly following acute brain injury to reduce cerebral edema (Cook et al., 2020; DeHoff and Lau, 2022; Gharizadeh et al., 2022; Schwarz et al., 2002; Shi et al., 2020; Surani et al., 2015), but also as a part of treatment for cystic fibrosis (Elkins and Bye, 2011), asthma (Daviskas et al., 1996), bronchiectasis (Kellett et al., 2005), COVID-19 (Gennari-Felipe et al., 2022), and resuscitation of critically ill patients (Pfortmueller and Schefold, 2017). Further, treatment with hyperosmotic solutions has been shown to limit inflammatory disease in preclinical disease models, including kainite-induced brain injury (Compan et al., 2012), sepsis (Theobaldo et al., 2012), intracerebral hemorrhage (Schreibman et al., 2018), pulmonary injury due to ischemia–reperfusion (Shields et al., 2003), and ARDS (Petroni et al., 2015), suggesting that manipulation of cell volume regulation could be harnessed to target disease processes. This is reflected in our study where hyperosmotic solutions were capable of impairing PAMP-induced IL-6 release, as well as limiting cell swelling–induced type I IFN production and inflammasome activation. Future studies will reveal the potential for regulating VRAC-dependent RVD and cell volume changes in macrophages in disease.

## Materials and methods
### Animals and cell culture
Conditional CX3CR1-driven macrophage-specific VRAC KO mice (*Cx3cr1[Cre/+];Lrrc8a[flx/flx]*) with WT littermates (*Cx3cr1[+/+]; Lrrc8a[flx/flx]*) were generated and maintained at the University of Manchester as previously described (Cook et al., 2022; Green et al., 2020). IFNAR KO mice (Muller et al., 1994) (B6.129S2-Ifnar1[tm1Agt/Crs]) were originally provided by C. Reis e Sousa (The Francis Crick Institute, London, UK). IFNAR KO mice with WT littermates were bred and provided locally by Prof Joanne Konkel (University of Manchester, Manchester, UK). Animals were housed in ventilated cages with temperature and humidity maintained between 20 and 24°C and 45 and 65%, respectively, with a 12-h light–dark cycle. All procedures were performed with appropriate personal and project licenses in place, in accordance with the Home Office (Animals) Scientific Procedures Act (1986), approved by the Home Office and the local Animal Ethical Review Group, University of Manchester, and reported according to the ARRIVE guidelines. Primary BMDMs were generated using bone marrow isolated from femurs and tibias from mice described above. In brief, bone marrow was collected, erythrocytes were lysed, and subsequently isolated bone marrow was cultured in DMEM (10% vol/vol FBS, 100 U ml[−1] penicillin, 100 μg ml[−1] streptomycin) supplemented with L929-conditioned media (30% vol/vol) for 6–7 days. BMDM cultures were fed with extra media (containing 30%

vol/vol L929-conditioned media) on day 3. Before experiments, BMDMs were scraped and seeded overnight at a density of 1 × 10[6] ml[−1] in DMEM (10% vol/vol FBS, 100 U ml[−1] streptomycin/penicillin). Mouse L929 fibroblasts were cultured in DMEM (10% vol/vol FBS, 100 U ml[−1] penicillin, 100 μg ml[−1] streptomycin). L929 cells were disassociated with trypsin and seeded overnight at a density of 1 × 10[6] ml[−1].

### Cell stimulation
For experiments where cell volume changes were induced by hypo-osmotic shock, BMDMs were incubated in iso-osmotic DMEM (50% vol/vol serum-free DMEM, 50% vol/vol sterile phosphate-buffered saline (PBS) [no $Mg^{2+}$/$Ca^{2+}$]) or hypo-osmotic DMEM (50% vol/vol serum-free DMEM, 50% vol/vol sterile, low endotoxin, cell culture grade, $H_2O$). Therefore, hypo-osmotic DMEM had an osmolarity of ~170 mOsm and iso-osmotic DMEM had an osmolarity of ~310 mOsm.

For experiments inhibiting TBK1/IKKε, BMDMs were treated with MRT67307 (10 μM) or vehicle control (DMSO, 0.1% vol/vol). For experiments inhibiting lysosomal pH (and thus endolysosomal TLR signaling), BMDMs were treated with BafA1(100 nM) or vehicle control (DMSO, 0.5% vol/vol). For experiments inhibiting the cGAS-STING pathway, BMDMs were treated with either a vehicle control (DMSO, 0.5% vol/vol), H151 (10 μM), A151 (1 μM), Ru.251 (10 μM), 4-sulfonic calix[6]arene (30 μM), NU7441 (100 nM), or brefeldin A (10 μg ml[−1]). All compounds were added for 15 min prior to the addition of PBS (50% vol/vol) or $H_2O$ (50% vol/vol), also containing inhibitors. For experiments with chemical activation of cGAS-STING, BMDMs were incubated in serum-free DMEM and stimulated with CMA (250 μg ml[−1]) or Lipofectamine 3000–mediated transfection of poly dA:dT (1 μg ml[−1]) for 6 h. For experiments testing potentiation of CMA-induced STING signaling by cell volume, CMA was added immediately after incubation in iso-osmotic (50% PBS vol/vol in DMEM) or hypo-osmotic (50% $H_2O$ vol/vol in DMEM) media for 3 h (pTBK1 and pIRF3), 4 h (STING dimerization), or 6 h (qRT-PCR and ELISA). For experiments testing 2′3′-cGAMP transport, BMDMs and L929 fibroblasts were incubated in iso-osmotic or hypo-osmotic DMEM and incubated with extracellular 2′3′-cGAMP (5 μg ml[−1]) for 3 h. For experiments testing membrane disruption using Lipofectamine, BMDMs were incubated with the membrane-stabilizing agent punicalagin (Martin-Sanchez et al., 2016) (50 μM) for 15 min before stimulation with Lipofectamine 3000 (0.5% vol/vol) and incubated for 6 h.

For experiments investigating TLR responses, BMDMs were incubated in either serum-free DMEM, or DMEM with osmolarity altered as follows: hypo-osmotic DMEM (50% vol/vol $H_2O$ in DMEM; ~170 mOsm), iso-osmotic DMEM (50% vol/vol 150 mM NaCl in DMEM; ~320 mOsm), and hyperosmotic DMEM (50% vol/vol 300 mM NaCl in DMEM; ~470 mOsm). In parallel, cells were stimulated by the addition of either poly I:C (5 μg ml[−1]), LPS (1 μg ml[−1]), imiquimod (25 μM), or CpG-DNA ODN-1826 (5 μg ml[−1]) for 6 h.

### RNAi knockdown in BMDMs
On day 6 after isolation, WT and VRAC KO BMDMs were seeded out at 0.5 × 10[6] ml[−1] in 12-well plates in DMEM (10% vol/vol FBS,

100 U ml⁻¹ penicillin, 100 µg ml⁻¹ streptomycin) supplemented with L929-conditioned media (30% vol/vol). After overnight incubation, BMDMs were treated with SMARTpool ONTarget PLUS siRNAs (Dharmacon, Horizon Discovery) targeting *Sting1* (L-055528-00-0005), *Cgas* (L-055608-01-0005), *Mavs* (L-053767-00-0005), or nontargeting control (D-001810-10-05) at a concentration of 75 nM using Lipofectamine RNAiMAX (3 µl/well) according to the manufacturer's protocol, for 48 h. The media were replaced with fresh DMEM (10% vol/vol FBS, 100 U ml⁻¹ penicillin, 100 µg ml⁻¹ streptomycin) the evening before stimulation with iso- or hypo-osmotic media for 6 h.

## RNA sequencing
RNA was isolated using a PureLink RNA Miniprep kit following the manufacturer's instructions. Total isolated RNA was submitted to the Genomic Technologies Core Facility at the University of Manchester. RNA samples were assessed for quality and integrity using a 2200 TapeStation (Agilent Technologies), and then, libraries were generated using the TruSeq Stranded mRNA assay (Illumina, Inc.) according to the manufacturer's protocol. Briefly, polyadenylated mRNA was purified using poly-T, oligo-attached magnetic beads, fragmented using divalent cations under elevated temperature, and then reverse-transcribed into first-strand cDNA using random primers. Second-strand cDNA was then synthesized using DNA Polymerase I and RNase H. Following a single "A" base addition, adapters were ligated to the cDNA fragments, and the products were then purified and enriched by PCR to create the final cDNA library. Adapter indices were used to multiplex libraries, which were pooled prior to cluster generation using a cBot instrument. The loaded flow cell was then paired-end–sequenced (76 + 76 cycles, plus indices) on an Illumina HiSeq 4000 instrument. Finally, the output data were demultiplexed (allowing one mismatch) and BCL-to-Fastq conversion performed using Illumina's bcl2fastq software, version 2.20.0.422.

Raw data have been deposited to BioStudies ArrayExpress and are available using accession number E-MTAB-16723. Normalized transcript data from the RNA-seq can be found in Data S1. It is important to note that in the RNA-seq dataset, *Lrrc8a* RNA was still detected in KO macrophages, likely since only exon 3 of the *Lrrc8a* gene was deleted. Our previous study (Cook et al., 2022) revealed using exon-specific primers intact expression of exons 1–2, but complete loss of exon 3 transcripts, in VRAC KO cells from *Lrrc8a*^fl/fl^ *Cx3cr1*^cre/+^ mice.

For analysis, unmapped paired-end sequences were tested by FastQC (https://www.bioinformatics.babraham.ac.uk/projects/fastqc/). Sequence adapters were removed, and reads were quality-trimmed using Trimmomatic_0.36 (Bolger et al., 2014). The reads were mapped against the reference mouse genome (mm10/GRCm38), and counts per gene were calculated using annotation from GENCODE M21 (https://www.gencodegenes.org/) using STAR_2.5.3a (Dobin et al., 2013). Normalization, principal components analysis, and differential expression were calculated with DESeq2_1.28.1 (Love et al., 2014). Log2 fold change values for each comparison were calculated in DESeq2 using the standard negative binomial model and the apeglm method for shrinkage estimation. For comparisons between

hypotonic- and isotonic-treated cells within the same genotype, paired analysis was used to correct for within-group expression variability at baseline. DEGs between conditions were defined using thresholds of Log2 fold change >0.5, adjusted P value <0.01.

Gene expression data were analyzed in R (version 4.4.1) using the ClusterProfiler and msigdbr packages for GSEA enrichment. The resulting data were processed using Cytoscape with the EnrichmentMap plugin to produce enrichment maps of the identified gene sets.

## qRT-PCR
RNA was isolated using a PureLink RNA Miniprep kit following the manufacturer's instructions. cDNA was generated from isolated RNA using SuperScript III first-strand synthesis kits. qRT-PCR was performed using 5 ng cDNA, 200 nM forward and reverse primers, and SYBR Green Mastermix according to the manufacturer's instructions. Primer sequences used were as follows: *Ifnb* FWD: 5′-TGGGAGATGTCCTCAACTGC-3′, REV: 5′-CCAGGCGTAGCTGTTGTACT-3′; *Cxcl10* FWD: 5′-CCACGTGTTGAGATCATTGCC-3′, REV: 5′-TCACTCCAGTTAAGGAGCCC-3′; *Rsad2* FWD: 5′-TGGTTCAAGGACTATGGGGAGT-3′, REV: 5′-CTTGACCACGGCCAATCAGA-3′; *Ccl7* FWD: 5′-CCCTGGGAAGCTGTTATCTTCAA-3′, REV: 5′-CTCGACCCACTTCTGATGGG-3′; *Il1b* FWD: 5′-CCACAGACCTTCCAGGAGAATG-3′, REV: GTGCAGTTCAGTGATCGTACAGG-3′; *Ifna* (pan-isoforms) FWD: 5′-TTTCCCCTGACCCAGGAAGA-3′, REV: 5′-GGCTCTCCAGACTTCTGCTC-3′; Il6 FWD: 5′-CTCTGGGAAATCGTGGAAAT-3′, REV: 5′-CCAGTTTGGTAGCATCCATC-3′; Tnf FWD: 5′-CCCTCACACTCAGATCATCTTCT-3′, REV: 5′-GCTACGACGTGGGCTACAG-3′; Il18 FWD: 5′-GACTCTTGCGTCAACTTCAAGG-3′, REV: 5′-CAGGCTGTCTTTTGTCAACGA-3′; Cgas FWD: 5′-CAGAAACGGGAGTCGGAGTT-3′, REV: 5′-TGTAGCTCAATCCTGGGGACT-3′; Mavs FWD: 5′-TCTCCTAACCAGCAGGCTCT-3′, REV: 5′-AACGGTTGGAGACACAGGTC-3′; Sting1 FWD: 5′-GTCTAGGAAGCAGAAGATGCCA-3′, REV: 5′-GCTGGCCACCAGAAAGATGA-3′; IAV M1 FWD: 5′-AAGACCAATCCTGTCACCTCTGA-3′, REV: 5′-CAAAGCGTCTACGCTGCAGTCC-3′; IAV M2 FWD: 5′-CAGGTCGAAACGCCTATCAG-3′, REV: 5′-CAAGTGCAAGATCCCAATGA-3′; Lrrc8a (exon 3) FWD: 5′-ACATCCCCGACGTCAAGAAC-3′, REV: 5′-GCGCAGCTTGTTTTCACTCA-3′; Gapdh FWD: 5′-CAGTGCCAGCCTCGTCC-3′, REV: 5′-CAATCTCCACTTTGCCACTGC-3′. All samples were loaded in triplicate and assayed on a 7900HT Fast Real-Time PCR machine (Applied Biosystems) for 40 cycles with standard settings. For each primer pair, a standard curve consisting of four 10-fold dilutions of neat cDNA was run in parallel to determine amplification efficiency. Sample-wise abundance was normalized to GAPDH to correct for loading differences, then further normalized to WT isotonic-treated cells to give fold change values.

## Western blotting
BMDM lysates were generated by lysis in Triton X-100 lysis buffer (50 mM Tris-HCl, 150 mM NaCl, 1% vol/vol Triton X-100, pH 7.3) containing protease inhibitor cocktail. Lysates were mixed with 5× Laemmli buffer and boiled at 95°C for 5 min. Lysates were separated by Tris-glycine SDS-PAGE and

transferred onto nitrocellulose or PVDF membranes using a semi-dry Trans-Blot Turbo system. Membranes were blocked in 5% wt/vol milk in PBS, 1% vol/vol Tween-20 (PBS-T) before being incubated with indicated antibodies (mouse anti-viperin, clone MaP.VIP, catalog#MABF106; Millipore; rabbit anti-phospho-IRF-3 (Ser396), clone 4D4G, catalog#4947; CST; rabbit anti-IRF3, clone D83B9, catalog#4302; CST; rabbit anti-phospho-TBK1/NAK (Ser172), clone D52C2, catalog#5483; CST; rabbit anti-TBK1/NAK, clone D1B4, catalog#3504; CST; goat anti-mouse IL-1β, catalog#AF-401; R&D; mouse anti-LRRC8A, clone 8H9, catalog#sc-517113; Santa Cruz; rabbit anti-caspase-3, catalog#9662; CST; rabbit anti-caspase-1, clone EPR16883, catalog#ab179515; Abcam) in 5% wt/vol BSA in PBS-T, overnight at 4°C. Membranes were washed three times before incubation with appropriate HRP-tagged secondary antibodies (1 h, RT), washed three times with PBS-T, and then visualized with Amersham ECL Prime detection reagent (GE Healthcare). For phospho-blots of IRF3 and TBK1, membranes were blocked in 5% BSA (wt/vol), 1% Tween-20 (vol/vol) in Tris-buffered saline (TBS-T) and washed with TBS-T. Chemiluminescence was visualized using a G:Box Chemi XX6 (Syngene). Membranes were reprobed for β-actin using HRP-conjugated mouse anti-β-actin antibodies (clone AC-15, catalog#A5441; Sigma-Aldrich).

For STING dimerization assays, BMDMs were lysed in NP-40 lysis buffer (50 mM NaCl, 50 mM Tris-HCl, pH 7.4, 2 mM EDTA, 0.5% NP-40 vol/vol) for 1 h. Lysates were clarified by centrifugation ($10,000 \times g$, 10 min, 4°C) and then mixed with 5× nondenaturing Laemmli buffer (300 mM Tris, pH 6.8, 50% vol/vol glycerol, 10% wt/vol SDS, 0.25% wt/vol bromophenol blue) and incubated for 10 min at 37°C. Lysates were then run by SDS-PAGE as described above, using rabbit anti-STING antibodies (clone D2P2F, catalog#13647; CST).

For puromycin incorporation assays to assess translation, BMDMs were incubated with 1 µg ml$^{-1}$ puromycin for 30 min prior to the end of the experiment. BMDMs were lysed in RIPA buffer (50 mM NaCl, 50 mM Tris-HCl, pH 7.4, 1% NP-40 vol/vol, 0.1% SDS wt/vol, 0.5% sodium deoxycholate wt/vol) and run by SDS-PAGE as above using anti-puromycin antibodies (clone 12D10, catalog#MABE343; Merck).

### Cell death assays
Cell death was assessed via a MTT assay kit (ab211091; Abcam), an LDH release kit (G1780; Promega), and a Caspase-Glo 3/7 kit (G8090; Promega) according to the manufacturer's instructions. For MTT and LDH, $0.1 \times 10^6$ BMDMs were incubated for 6 h in iso-osmotic (50% PBS vol/vol in DMEM) or hypo-osmotic (50% $H_2O$ vol/vol in DMEM) media before the supernatant was removed and used for LDH assay (performed according to the manufacturer's instructions). Cells were then placed in 100 µl 50% vol/vol MTT assay mix in Opti-MEM for 3 h at 37°C before addition of 150 µl MTT solvent, mixed for 1 h at RT, and read at A590. For Caspase-Glo 3/7 assays, BMDMs were seeded in white 96-well plates and stimulated as above. After stimulation, 50 µl of supernatant was removed and replaced with Caspase-Glo 3/7 reagent, briefly mixed, and incubated at RT for 1 h before luminescence was read. Where indicated, BMDMs were pretreated for 15 min with either STING inhibitors (H151, 10 µM),

necroptosis inhibitors (RIPK1 inhibitor necrostatin-1, 50 µM), pan-caspase inhibitors (ZVAD-FMK, 50 µM), or caspase-1 inhibitors (VX765, 10 µM).

### Immunofluorescence
For immunostaining, WT and VRAC KO BMDMs were adhered to sterile 13-mm glass coverslips at a density of $1 \times 10^6$ ml$^{-1}$. After overnight incubation, cells were stimulated as described. Cells were then washed once with cold PBS ($+Ca^{2+}/Mg^{2+}$) and fixed with 4% PFA (wt/vol) for 10 min. Fixed cells were washed three times with PBS and blocked and permeabilized with 1% BSA (wt/vol), 0.1% Triton X-100 (vol/vol) in PBS for 1 h. Cells were stained anti-G3BP1 (clone 1E4A2, Proteintech) or galectin-3 (clone M3/38, Santa Cruz) in blocking buffer overnight at 4°C. Coverslips were then washed three times with PBS before incubation with anti-rabbit or anti-mouse Alexa 488–conjugated secondary antibodies for 1 h at RT. After three washes with PBS, cells were stained with DAPI (1 µg ml$^{-1}$) and washed a further three times with dH$_2$O before mounting in Prolong Gold (Thermo Fisher Scientific). Images were captured using a ×63/1.4 Plan Apochromat objective on a Zeiss Axio Imager D2 upright microscope and captured using a CoolSNAP HQ2 camera (Photometrics) through Micro-Manager software v1.4.23. Images were subsequently processed and analyzed using ImageJ software.

### IAV infection model
IAV (H1N1) strain A/PR8/8/34 (VR-1469; ATCC) was used to model IAV infection in BMDMs. IAV stocks were propagated in MDCK cells to a PFU between $1 \times 10^7$ and $1 \times 10^8$. WT and VRAC KO BMDMs were seeded out at $0.5 \times 10^6$ ml$^{-1}$ in 12-well plates in DMEM (10% vol/vol FBS, 100 U ml$^{-1}$ streptomycin/penicillin). BMDMs were washed with serum-free DMEM before infection with IAV (MOI 5 or 10 as indicated in DMEM (1% vol/vol FBS, antibiotic free)) for 2 h, shaken every 30 min. The media were then removed and replaced with DMEM (10% FBS vol/vol) and incubated for the indicated time points. Cell supernatants were collected for CXCL10 ELISA (DY466-05; R&D Systems, according to the manufacturer's instructions), and RNA was isolated from BMDMs using a PureLink RNA Miniprep kit following the manufacturer's instructions.

### *In vivo* hyperinflammation model
A mouse model of CpG-induced MAS was performed as previously described (Behrens et al., 2011; Gleeson et al., 2024). Conditional CX3CR1-driven macrophage-specific VRAC KO mice (*Cx3cr1*$^{Cre/+}$;*Lrrc8a*$^{flx/flx}$) and WT littermates (*Cx3cr1*$^{+/+}$; *Lrrc8a*$^{flx/-flx}$) were used, aged 16 wk and of both sexes. To induce MAS, mice received five i.p. injections of CpG ODN 1826 oligodeoxynucleotides (5′-T*C*C*A*T*G*A*C*G*T*T*C*C*T*G*A*C*G*T*T-3′, where * indicates a phosphorothioate modification) (2 mg kg$^{-1}$) or vehicle control (sterile PBS, 10 µl g$^{-1}$) over 10 days. Mice were randomly assigned to treatment (WT PBS, $n = 5$; KO PBS, $n = 5$; WT CpG, $n = 6$; KO CpG, $n = 7$), and researchers were blinded to both genotype and treatment for the duration of the experiment and analyses. Mice received one i.p. injection of CpG or PBS on days 0, 2, 4, 7, and 9, with tissue collected on day 10, 24 h after the last injection. Animals were weighed daily. Mice were anesthetized,

blood was collected via cardiac puncture, and plasma was isolated by spinning at 1500 × g for 15 min at 4°C, before storage at –70°C. Following cardiac puncture, animals were transcardially perfused with PBS before collection and weighing of the spleen and liver. Plasma ferritin levels were assessed via ELISA (ab157713; Abcam), plasma IL-18 levels were assessed via ELISA (BMS618-3; Invitrogen), and plasma IFNγ, TNF, IL-6, and IL-10 were determined by a LEGENDplex multiplex assay (mouse inflammation panel, 740446; BioLegend), all according to the manufacturer's instructions.

### RVD assay

$5 \times 10^4$ BMDMs were seeded out into black-walled 96-well plates overnight. Cells were loaded with 10 μM calcein AM (10 μM, 1 h, 37°C), washed three times with DMEM, and then rested for 30 min to allow calcein to equilibrate. After three further washes with serum-free DMEM, cells were imaged on an Eclipse Ti inverted microscope (Nikon), using a 40×/0.6 objective SPlan Fluor objective, equipped with a stage incubator maintaining 37°C and 5% $CO_2$. Calcein fluorescence was captured in the GFP channel using low laser power to minimize phototoxicity and bleaching. Imaging software NIS-Elements AR.46.00.0 was used for acquisition. Live imaging was conducted using point-visiting to image all conditions simultaneously. One image was captured every 2 min, with hypotonic shock induced at 5 min by adding endotoxin-free distilled $H_2O$ (50% vol/vol) for a final osmolarity of ~170 mOsm $kg^{-1}$. For quantification, fluorescent images were processed in Fiji, using a rolling-ball background subtraction (50-pixel ball size) to remove non–cell-associated fluorescence. The average pixel intensity of each frame was then measured and normalized to the value of the first frame, yielding $F/F_0$ curves. For statistical testing, AUC values of the curves were calculated in Prism.

### Quantification and statistical analysis

Data are presented as mean values plus the SEM. Accepted levels of significance were $*P < 0.05$, $**P < 0.01$, $***P < 0.001$. Statistical analyses were carried out using GraphPad Prism (version 9). Equal variance and normality were assessed with Levene's test and the Shapiro–Wilk test, respectively, and appropriate transformations were applied. Groups containing normally distributed data were analyzed using either a one-way ANOVA with Sidak's or Tukey's post hoc analysis, or a two-way ANOVA with Sidak's, Tukey's, or Dunnett's post hoc analysis as appropriate. n represents experiments performed on individual animals or cells acquired from individual animals.

### Online supplemental material

Fig. S1 (related to Fig. 1) shows quantification of RVD in WT and VRAC BMDMs used for RNA-seq analysis, and GSEA between WT and VRAC KO BMDMs. Fig. S2 (related to Fig. 2) shows additional cytokines influenced by cell volume (*Il18, Ifna, Tnf*) and validation of a lack of IFN signaling in IFNAR KO BMDMs. Fig. S3 (related to Fig. 3) shows hypo-osmotic type I IFN signaling functions independently of DNA-PK and validation of the DNA sensing and STING inhibitors used in this study. Fig. S4 shows representative time-lapse microscopy images of WT and VRAC

KO BMDMs incubated in iso- or hypo-osmotic media for 4 h. Fig. S5 shows that there is no impact on lysosomal integrity following hypo-osmotic medium treatment in WT or VRAC KO BMDMs via galectin-3 staining and DQ-BSA uptake. Data S1 shows the normalized transcript counts for each RNA transcript from the RNA-seq analysis of WT and VRAC KO BMDMs incubated in iso- or hypo-osmotic media for 4 h. Table S1 shows the GSEA reports of upregulated and downregulated gene sets in WT BMDMs incubated from RNA-seq analysis.

## Data availability

The data are available from the corresponding author upon reasonable request. Raw data from RNA-seq datasets can be found on BioStudies ArrayExpress, accession number E-MTAB-16723.

## Acknowledgments

We thank Professor Joanne Konkel (Faculty of Biology of Medicine and Health, Lydia Becker Insitute of Immunology and Inflammation, University of Manchester, UK) for sharing IFNAR KO mice, originally provided by C. Reis e Sousa lab (Immunobiology Laboratory, Francis Crick Institute, UK). We would additionally like to thank the University of Manchester Genomic Technologies Core Facility, and Leo Zeef in the Bioinformatics Core Facility for their assistance with RNA-seq. The Bioimaging Facility microscopes used in this study were purchased with grants from BBSRC, Wellcome, and the University of Manchester Strategic Fund.

This study was supported by grants from the Medical Research Council, UK (MRC, MR/W028867/1, MR/T016515/1, MR/N013751/1). T.A. Gleeson received studentship funding from the MRC (MR/R015767/1) and the Swedish Orphan Biovitrum (Sobi). J.P. Green received funding from a Presidential Fellowship from the University of Manchester. Open Access funding provided by University of Manchester.

Author contributions: James R. Cook: conceptualization, formal analysis, investigation, methodology, and writing—review and editing. Tara A. Gleeson: formal analysis and investigation. Sara Gago: methodology, resources, and writing—review and editing. Stuart M. Allan: writing—review and editing. Kevin N. Couper: funding acquisition, supervision, and writing—review and editing. Catherine B. Lawrence: conceptualization, funding acquisition, supervision, and writing—review and editing. David Brough: conceptualization, funding acquisition, methodology, project administration, resources, supervision, visualization, and writing—review and editing. Jack P. Green: conceptualization, formal analysis, funding acquisition, investigation, project administration, supervision, visualization, and writing—original draft, review, and editing.

Disclosures: The authors declare no competing interests exist.

Submitted: 19 November 2024

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

# Supplemental material

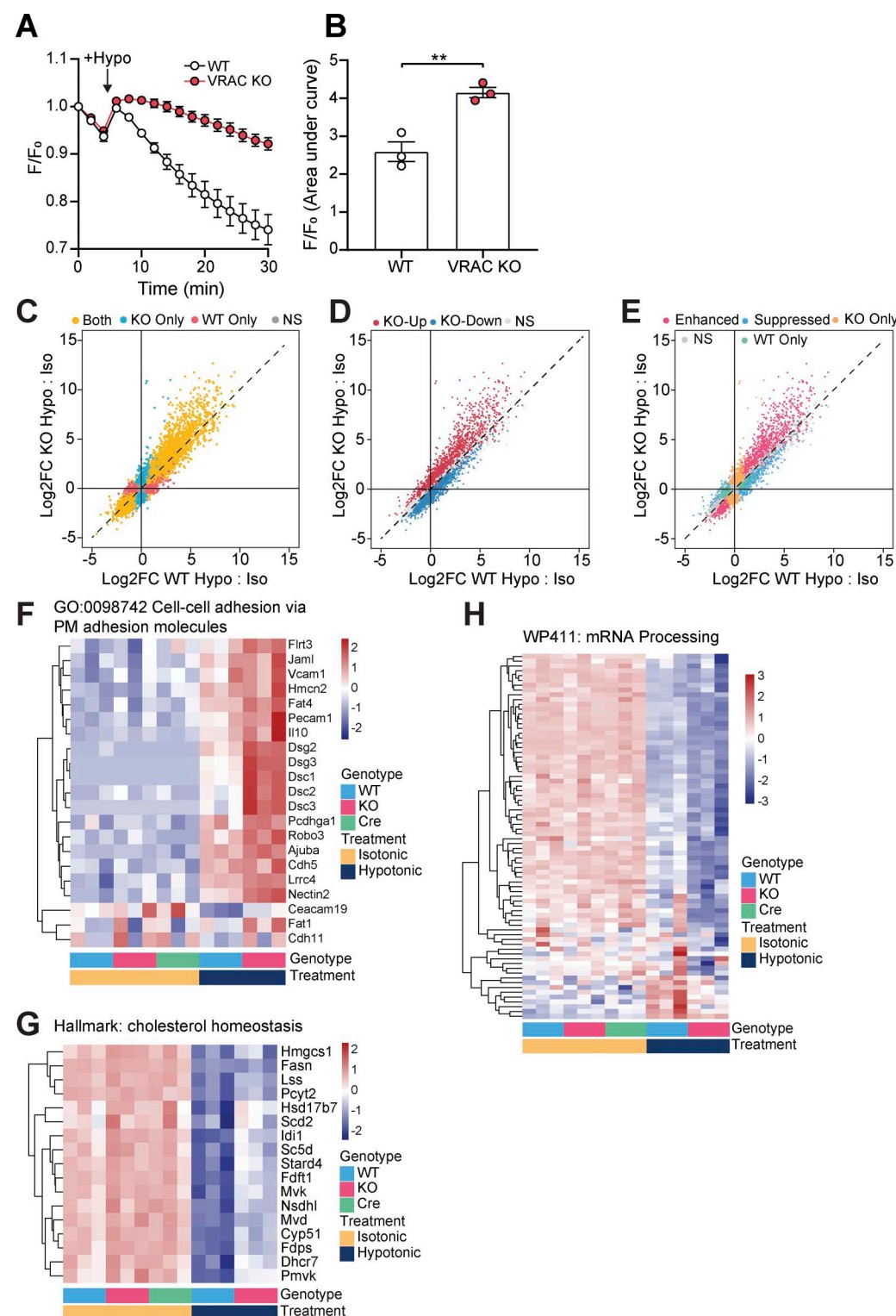

Figure S1. **Related to Fig. 1: KO of LRRC8A removes RVD in BMDMs and further RNA-seq analysis. (A)** RVD measured by calcein fluorescence in WT or VRAC KO BMDMs incubated in hypo-osmotic media (50% vol/vol $H_2O$ in DMEM, ~170 mOsm $kg^{-1}$), added at the indicated time point ($n$ = 3). These data were obtained from the same cultures of BMDMs used for RNA-seq analysis in Fig. 1. **(B)** Area under the curve analysis of A. **(C–E)** Scatter plot comparing gene-wise Log2FC values following hypotonic treatment in WT and KO BMDMs, labeled by significance compared with respective isotonic controls (C), by significance in KO cells compared with WT under hypotonic conditions (D), and by the effect of VRAC KO on modifying the WT hypotonic effect (E). **(F–H)** Heatmaps displaying expression of the core enrichment genes (determined via GSEA) from gene sets differentially regulated in KO vs WT cells following hypotonic treatment, scaled by gene-wise Z-score. **P < 0.01, determined by an unpaired $t$ test. Values shown are the mean ± the SEM. Log2FC, Log2 fold change.

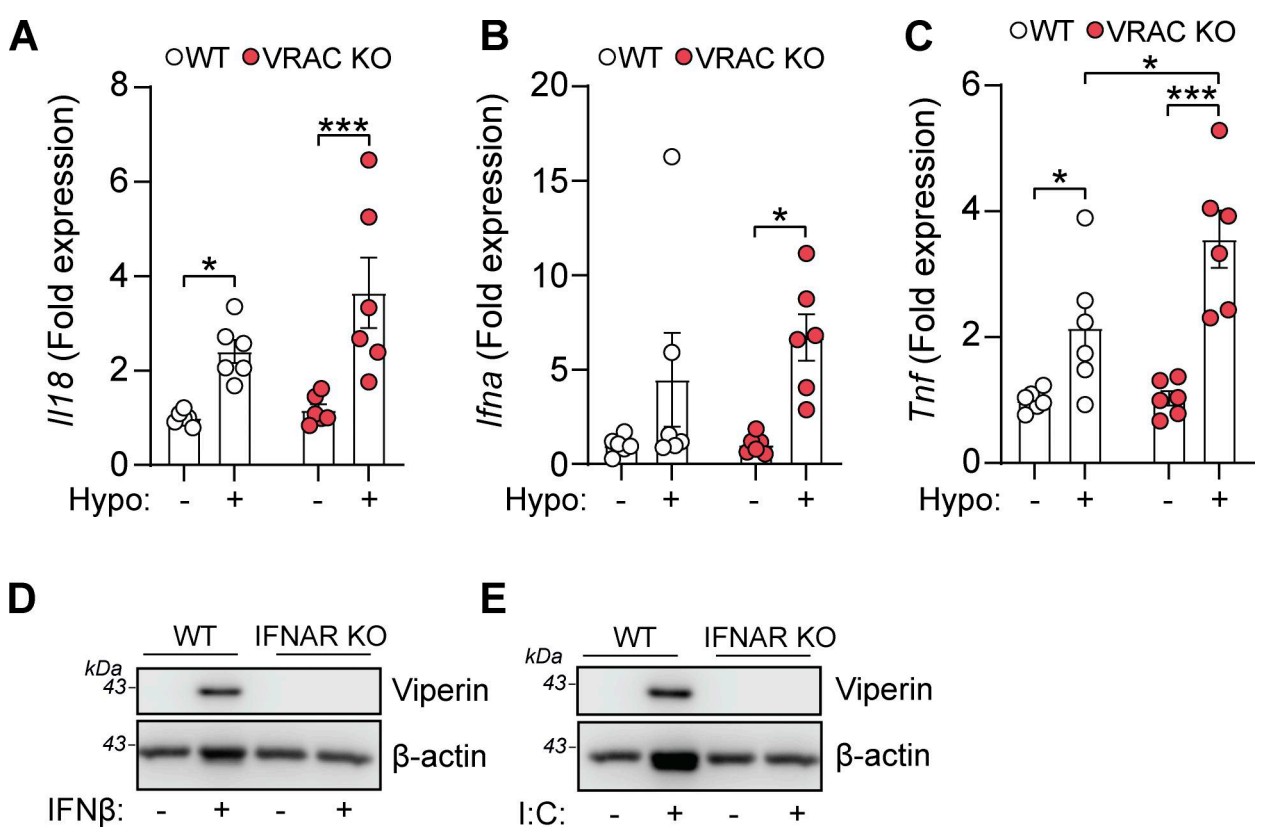

Figure S2.　**Related to Fig. 2: additional cytokines influenced by cell volume and validation of IFNAR KO BMDMs. (A–C)** qRT-PCR analysis of *Il18* (A), pan-Ifna transcripts (B), and *Tnf* (C) in WT and VRAC KO BMDMs incubated in iso- or hypo-osmotic media (6 h) (*n* = 6). **(D and E)** Western blot for viperin in WT and VRAC KO BMDMs stimulated with recombinant murine IFNβ (200 U/ml) (D) or poly I:C (1 μg/ml) (E) (6 h) (*n* = 3). *P < 0.05, ***P < 0.001, determined by a two-way ANOVA with Dunnett's post hoc analysis. Values shown are the mean ± the SEM.

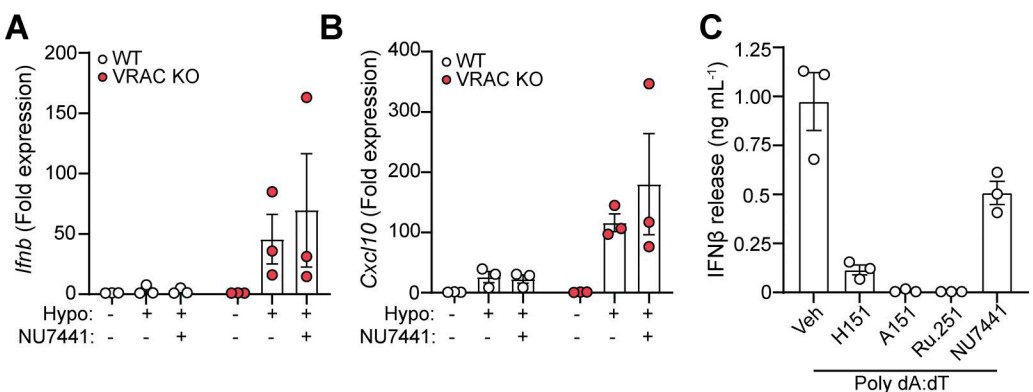

Figure S3.　**Related to Fig. 3: changes in cell volume drive IFNβ responses through a DNA- and TBK1-dependent pathway. (A and B)** qRT-PCR analysis of *Ifnb* (A) and *Cxcl10* (B) in WT and VRAC KO BMDMs incubated in hypo-osmotic media (50% $H_2O$ vol/vol in DMEM) for 6 h in the presence of Nu7441 (100 nM) or vehicle control (DMSO 0.5% vol/vol) (*n* = 3). **(C)** IFNβ release in supernatant from WT BMDMs stimulated with transfected poly dA:dT (1 μg/ml) in the presence of H151 (10 μM), A151 (1 μM), Ru.251 (10 μM), NU7441 (100 nM), or vehicle control (DMSO, 0.5% vol/vol) for 6 h (*n* = 3). Values shown are the mean ± the SEM.

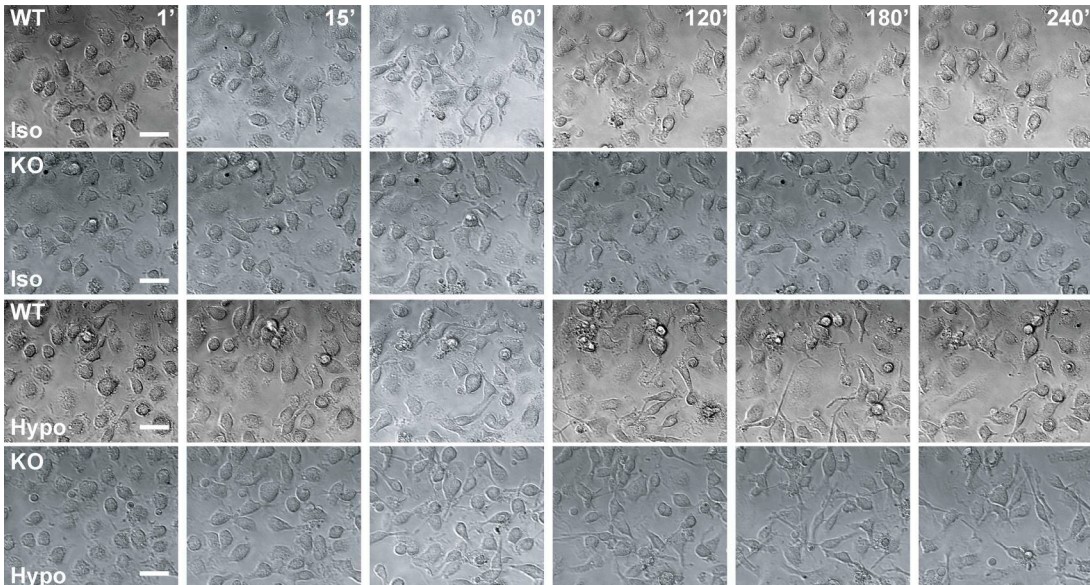

Figure S4.    **Time-lapse bright-field imaging of WT and VRAC KO BMDMs in hypo-osmotic media.** Representative images of WT and VRAC KO BMDM incubated in iso- or hypo-osmotic media. BMDMs were placed immediately in iso- or hypo-osmotic media immediately before commencing time-lapse microscopy ($n$ = 5) (scale = 25 μm).

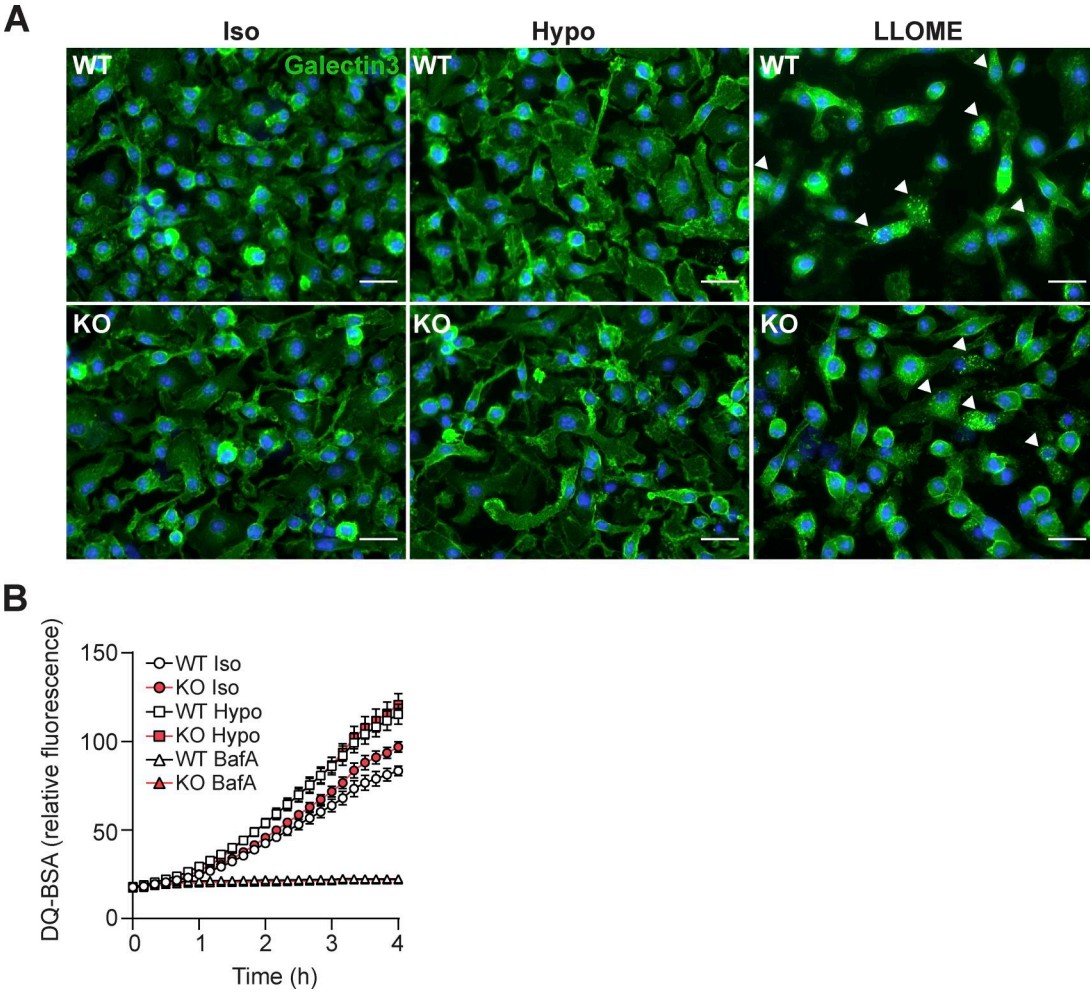

**Figure S5.   Lysosomal integrity and function are unaffected by hypo-osmotic media and loss of VRAC. (A and B)** Immunofluorescence for galectin-3 in WT and VRAC KO BMDMs incubated in iso- or hypo-osmotic media, or with the lysosomal permeabilization agent LLOME (1 mM) for 5 h (*n* = 3). Galectin-3 puncta–positive cells are highlighted with white arrows (scale = 25 µm) **(B)** Time-lapse of DQ-BSA fluorescence in WT and VRAC KO BMDMs incubated in iso- or hypo-osmotic media (*n* = 4). BafA1 (100 nM) was included as a control of reduced lysosomal function. Values shown are the mean ± the SEM.

**Provided online are Table S1 and Data S1. Table S1 shows the GSEA reports of upregulated and downregulated gene sets in WT BMDMs incubated from RNA-seq analysis. Data S1 shows the normalized transcript counts for each RNA transcript from the RNA-seq analysis of WT and VRAC KO BMDMs incubated in iso- or hypo-osmotic media for 4 h.**

