## [Peer Review File · The Journal of Cell Biology]

Disruption of macrophage cell volume drives inflammatory responses and type I interferon signalling

James Cook, Tara Gleeson, Sara Gago, Stuart Allan, Kevin Couper, Catherine Lawrence, David Brough, and Jack Green

Corresponding Author(s): Jack Green, University of Manchester

Review Timeline:

Submission Date:	2024-11-19
Editorial Decision:	2025-01-03
Revision Received:	2026-01-15
Editorial Decision:	2026-02-13
Revision Received:	2026-03-05

Monitoring Editor: Ana-María Lennon-Dumenil

Scientific Editor: Tim Spencer

Transaction Report:

DOI: <https://doi.org/10.1083/jcb.202411133>

January 3, 2025

Re: JCB manuscript #202411133

Jack Green
University of Manchester

Dear Dr. Green,

Thank you for submitting your manuscript entitled "Disruption of macrophage cell volume drives inflammation through type I interferon signalling". Your manuscript has been assessed by expert reviewers, whose comments are appended below. Although the reviewers express potential interest in this work, significant concerns unfortunately preclude publication of the current version of the manuscript in JCB.

As you will see, reviewers commended the clearly articulated findings that overall supported the main claims made in this work linking regulation of cell volume with inflammatory signaling in macrophages. However, in confidential comments to the editors as well as those shared with the authors, reviewers remarked that this work overlaps significantly with prior work by these authors. Concerning the relation to prior work, Reviewer 3 questioned the relevance of the "moderate" hypotonic swelling here vs the "severe" swelling examined previously for macrophage biology. We concur these issues diminish the novelty of the work and a revision should offer greater detail and validation of the involvement of STING signaling, IFN gene expression, and investigate more thoroughly the IL-18 increase seen in vivo. Reviewers 1 and 2 offer valuable guidance in sufficiently strengthening this work along these lines, and Reviewer 3 further suggests a potential model which the authors may consider. All specific points should be resolved in a suitable revision, which may required the authors to extend this from a Report into an Article.

Please let us know if you are able to address the major issues outlined above and wish to submit a revised manuscript to JCB. If so, before submitting a revision please send a revision plan with a point-by-point rebuttal to the reviewer comments. Only after approval of this plan will we consider a revised manuscript.

Note that a substantial amount of additional experimental data likely would be needed to satisfactorily address the concerns of the reviewers. The typical timeframe for revisions is three to four months. If you anticipate any difficulties in meeting this aforementioned revision time limit, please contact us and we can work with you to find an appropriate time frame for resubmission. Please note that papers are generally considered through only one revision cycle, so any revised manuscript will likely be either accepted or rejected.

If you choose to revise and resubmit your manuscript, please also attend to the following editorial points. Please direct any editorial questions to the journal office.

GENERAL GUIDELINES:

Text limits: Character count is < 40,000, not including spaces. Count includes title page, abstract, introduction, results, discussion, and acknowledgments. Count does not include materials and methods, figure legends, references, tables, or supplemental legends.

Figures: Your manuscript may have up to 10 main text figures. To avoid delays in production, figures must be prepared according to the policies outlined in our Instructions to Authors, under Data Presentation, <https://jcb.rupress.org/site/misc/ifora.xhtml>. All figures in accepted manuscripts will be screened prior to publication.

Supplemental information: There are strict limits on the allowable amount of supplemental data. Your manuscript may have up to 5 supplemental figures. Up to 10 supplemental videos or flash animations are allowed. A summary of all supplemental material should appear at the end of the Materials and methods section.

Please note that JCB now requires authors to submit Source Data used to generate figures containing gels and Western blots with all revised manuscripts. This Source Data consists of fully uncropped and unprocessed images for each gel/blot displayed in the main and supplemental figures. Since your paper includes cropped gel and/or blot images, please be sure to provide one Source Data file for each figure that contains gels and/or blots along with your revised manuscript files. File names for Source Data figures should be alphanumeric without any spaces or special characters (i.e., SourceDataF#, where F# refers to the associated main figure number or SourceDataFS# for those associated with Supplementary figures). The lanes of the gels/blots should be labeled as they are in the associated figure, the place where cropping was applied should be marked (with a box),

and molecular weight/size standards should be labeled wherever possible.

If you choose to resubmit, please include a cover letter addressing the reviewers' comments point by point. Please also highlight all changes in the text of the manuscript.

Regardless of how you choose to proceed, we hope that the comments below will prove constructive as your work progresses. We would be happy to discuss them further once you've had a chance to consider the points raised. You can contact the journal office with any questions at cellbio@rockefeller.edu.

Thank you for thinking of JCB as an appropriate place to publish your work.

Sincerely,

Ana-María Lennon-Dumenil
Monitoring Editor
Journal of Cell Biology

Tim Fessenden
Scientific Editor
Journal of Cell Biology

Reviewer #1 (Comments to the Authors (Required)):

The manuscript by Cook et al., describes how the volume regulated anion channel (VRAC) could contribute to control volume sensing in macrophages, resulting in signaling to inflammation. In the absence of the VRAC, a sustained increase in the cell volume leads to STING activation and interferon production. The manuscript is of very good scientific standards. Experiments are well controlled and well performed both in vivo and in vitro on primary murine macrophages.

Major comments

The title does not seem to reflect the results, as inflammation has not been demonstrated to be a consequence of the type I interferon signaling: "through type I interferon" should be rephrased, as there are no experiments to directly address this point (anti-IFNAR or IFNAR KO). In addition, the term 'disruption of cell volume' might be better replaced with a more precise expression, such as 'sustained increase in cell volume.'

The authors monitor increased cell volume but never show a picture of the cells in the various experimental conditions. Could they provide such images as additional figures?

How does the cell viability change during the experiments?

Are the osmolarity used in the experiments related to known in vivo pathological situations?

Figure 3: Are inflammatory cytokines other than CXCL10 modulated by VRAC ? Have the authors tested IL18 as in the in vivo situation?

Figure 5: The effect of the absence of VRAC on the inflammatory response is relatively minor and only significant for the secretion of IL-18. Is this related to the mode of secretion of this cytokine? Related to pore formation?

What dictated the choice of the in vivo experimental set up? Authors might have found a greater role for cell volume in immune responses in the context of a viral infection, where type I IFN plays a more prominent role? This could be better explained in the discussion, at least.

Minor comments

- Discussion, page 10: "Thus, in addition to Piezo1 and TRPV4, VRAC can also be considered a key mediator of mechano-immunity in macrophages, co-ordinating cell volume changes to translate disturbances in the inflammatory environment to tune

the immune response." The authors successfully demonstrated that VRAC is triggered by cell swelling and mediates a RVD in macrophages. However, the authors did not show that VRAC is responsible for the induction of an inflammatory response. This part of the discussion could be rewritten to make that point clearer to the reader.

- Figure 4I-N: In the legend of the figure and in the text, the authors only mention hypo-osmotic, isotonic or hyper-osmotic media, whereas they put the osmolarity values (170, 320 or 470 mOsm) in the figure itself. The correspondence between the two is found in the material and methods but should be clearer for the reader both in the figure legend and in the text.

- Discussion, page 9: "Tiered inflammatory responses related to inflammasome activation in response to increasing DAMP-mediated stress are proposed previously for mechanisms of IL-1 β release³⁵, and more recently for DNA sensing³⁶": "have been/were proposed previously"?

Reviewer #2 (Comments to the Authors (Required)):

Comments on "Disruption of macrophage cell volume drives inflammation through type I interferon signalling"

Cook and colleagues report interesting observations on the impact of cell volume regulation to the inflammatory activation of macrophage. The paper builds on previous observations from the group that the voltage regulated anion channel (VRAC) is important for NLRP3 activation upon cell swelling in hypotonic conditions. The authors now explore the response of macrophages to less extreme hypotonic stress and found that VRAC is important to prevent excessive inflammatory activation induced by cellular swelling and that this is mediated by STING activation but independent on cGAMP. The findings are interesting and timely as immune regulation by mechanotransduction is an exciting emerging field with implications in basic cell biology and potential translation impact.

The paper is generally solid and well-written, however, there are still important concerns that should be addressed.

As a step forward to the role of the inflammasome in harsh hypotonic conditions, the authors describe the involvement of STING in inducing type-I interferon signalling under mild hypotonic stress. The role of STING requires further experimental data, and the mechanism of STING activation in hypotonic conditions needs to be addressed.

Specific points:

Figure 1. The comments on the transcriptional data across the conditions are not exhaustive. Before focusing on inflammation, authors should extensively elaborate on the different pathways induced by hypo in WT and in VRAC and in WThypo vs VRACHypo. Interesting genes related to cytoskeletal remodelling and Rho signalling (Arhgap29, Dsg2, Dsg3) are not commented.

For data in Figure 2A-D, explain better in methods or legends how are the logFC2 values calculated. The finding that some genes are already slightly elevated in VRAC control should be commented.

In Figure 3, the inhibitors used have controversial and broad effects and are not sufficient to claim or exclude the different pathways. The analysis should be performed in STING or cGAS-depleted macrophages (CRISPR editing is feasible in BMDM) to make a point.

Second, measuring intracellular cGAMP and transfecting dsDNA into the cytosol would be important to understand if STING hyperactivation in hypo high is ligand-independent.

More critically: what is the mechanism behind enhanced STING activation upon swelling in VRAC ko? Is it enhanced dimerization, or increased ER to ERGIC translocation or as suggested in the discussion plasmamembrane disruption? The mechanism has to be experimentally addressed.

Given the previous data on inflammasome, it would be important to dose IL-1b upon STING blockade to understand the cross-talk/selectivity between the pathways.

In Figure 5: The increased cytokine storm after repeated CpG stimulation in VRAC is interesting. However, the link between macrophage activation and levels of plasma cytokine is very indirect. Data should be provided on macrophages in vivo to show the volume and the expression of some of the cytokines.

Minor points:

Enrichment map from GSEA of Fig.1F: It should be indicated in the figure that it derives from DEGs and also in the figure legend would be better to replace "least" and "most" genes per term with actual absolute values.

In Fig.3K-L the authors say that extracellular cGAMP does not induce a higher response in hypo-osmotic media, but this is not true as WT cells increased Ifnb1 as well as Cxcl10. In addition, data are completely masked by the high values of L929 cells. It would be important to plot each cell type in a separate graph and rephrase the text.

Fig.5E contains less animals compared to all the other figures on the panel. Authors should comment on why only in this figure animals are less since it derives from the same set of experiments I guess.

How osmolarity of media has been calculated? (i.e. the values of 170 mOsm for hypo-osmotic, 310 mOsm for iso-osmotic and

470 for hyper-osmotic). In the result section it is reported once 170 mOsm Kg-1 for a BMDM media, which kg-1 should not be there. Moreover, material and methods report iso-osmotic media as 310 mOsm, whereas in Fig.4 the iso-osmotic media is reported as 320 mOsm, this should be corrected.

List of DEGs (used to plot Fig.1C-E) should be provided in supplementary files as well as DEGs between WT and VRAC KO at steady state and a file containing data used to generated Fig.1F.

That " IL-18 is known to be induced by interferon", a reference should be added.

Reviewer #3 (Comments to the Authors (Required)):

Cook and colleagues present new evidence for the modulation of inflammatory signalling in macrophages upon disruption of cell volume. The study centers on the volume regulated anion channel (VRAC) which is made up of various subunits of LRRC8. VRAC restores cell volume on hypotonic shock by promoting the efflux of chloride and water from cells. Previous work has shown that disruption of cell volume by exposure to a hypotonic medium can instigate NLRP3 inflammasome activity when VRAC is missing. The authors state; however, that the previous work was done under extreme hypotonic conditions (117 mOsm Kg-1) and therefore choose to perform a new set of experiments under less severe hypotonic conditions (170 mOsm Kg-1) which they believe will result in less severe changes to cell volume. Less severe hypo-osmotic conditions led to large scale gene transcription changes with notable differences between VRAC sufficient and deficient cells in Type 1 IFN signalling-related genes. Some of these observations, initially done by RNAseq, were confirmed by rtqPCR and by western blotting. Interestingly, these transcription/translation changes appeared to at least partially involve the STING pathway as various inhibitors of the STING pathway reduced the effect. Next, the authors investigated whether VRAC-dependent cell volume control may impact inflammatory signalling by macrophages in response to various PAMPs. Loss of VRAC function did enhance type 1 IFN release and IL-6 release upon simultaneous exposure to hypo-osmotic medium and either poly(I:C), imiquimod, or CpG. The authors state that this is relevant in the physiological context since infection can lead to oedema, hypoxia, oxidative stress and metabolic dysfunction which can in turn modulate cell volume. Finally, the authors demonstrate that loss of VRAC function in macrophages can impact the cytokine milieu in an in vivo model of inflammation. This paper is well-written, and the data does appear to be in general agreement with the conclusions. I do have some concerns that are discussed below:

Minor concerns:

i) The authors use inhibitors to propose that a cGAS-independent pathway is induced by cell volume changes. Is there precedent for this? I would like to see this discussed in the discussion, specifically under what conditions has this been observed in the past and how does it relate to the present observation. This next part would be nice but, I want to be clear, is not required for revisions. Macrophages are notoriously good at pumping drugs/inhibitors out of the cell, is it possible to entertain a genetic model for confirming that hypo-osmotic medium drives a cGAS-independent pathway? I don't want to suggest generating KOs, but I see that Invivogen sells cGAS KO macrophage lines. I'm also not suggesting purchasing them (because as usual they are expensive!) but perhaps a quick e-mail to see if anyone at their present institute has these? Would STING signalling be intact upon exposure of those cells to hypo-osmotic medium?

ii) When discussing figure 3K-L, the authors state that the 3 hour time point is selected to limit the contribution of hypo-osmotic medium alone on *lfnb* expression. This seems at odds the actual data in figure 2D where *lfnb* expression is increased precisely between 2 and 4 hours. Yet, there is no *lfnb* response upon exposure to hypo-osmotic medium in VRAC KO cells in figure 3K. What accounts for this apparent inconsistency? Can the authors explain this in the text.

iii) There is some literature of VRAC/LRRC8 on the regulation of endosome swelling. I wonder if the authors can entertain the idea that loss of VRAC may in some way impact endosomal integrity and hence induce either inflammasome activity or the escape of poly(I:C)/LPS from endosomes. Indeed, macrophages are macropinocytic and constitutively form large endocytic organelles that will likely contain the poly(I:C) and LPS that the cells are bathed in. Loss of endosomal integrity has been linked to inflammasome activity as well. Could this, at least in part, be the basis of the observed synergy of VRAC-deficiency and inflammatory signalling? In an ideal world, it would be nice to see a galectin 3 stain in the VRAC-deficient macrophages that are exposed to hypo-osmotic medium. However, a brief mention of this possibility, if the authors agree that it is logical, in the discussion would suffice. It would help because in the current manuscript there is no mechanism for this synergy.

iv) Finally, I'm not sure how the in vivo work relates to the "less severe" hypo-osmotic conditions used throughout the paper. It does not really have the ability to discern the degree of cell swelling and in fact the only cytokine that is differentially modulated is IL-18 which is often associated with the NLRP3 inflammasome. So is this more consistent with "severe" swelling. Please discuss.

Dear Colleagues,

Many thanks for the excellent and constructive comments on our manuscript. We have addressed the comments and have provided a response detailed below. We believe the significant new experimental data provide important further mechanistic insight into how cell volume disruption drives inflammation in macrophages. All changes to the manuscript have been highlighted. Thank you again for your consideration,

Yours sincerely,

Jack Green

Reviewer(s)' Comments to Author:

Reviewer 1:

1.1) The title does not seem to reflect the results, as inflammation has not been demonstrated to be a consequence of the type I interferon signaling: "through type I interferon" should be rephrased, as there are no experiments to directly address this point (anti-IFNAR or IFNAR KO). In addition, the term 'disruption of cell volume' might be better replaced with a more precise expression, such as 'sustained increase in cell volume.'"

Response: We now provide data from BMDMs from IFNAR KO mice demonstrating that hypo-osmotic media induced inflammation is dependent on type I IFN signalling. Specifically, we show that mRNA induction of the ISGs *Cxcl10* and *Rsad2* (Fig 2K-L) and the translated protein viperin by hypo-osmotic media are absent in IFNAR KO BMDMs (Fig 2M). Other inflammatory cytokines, such as pro-IL1 β which is driven by NF- κ B, was however unaltered in IFNAR KO (Fig 2M), highlighting that not all inflammatory responses were driven by type I IFN signalling here. Specificity of IFNAR KO macrophages as deficient in type I IFN signalling is also demonstrated as they lacked viperin induction to recombinant IFN- β or Poly I:C (Supplementary Fig 2D-E). To reflect this, we have also amended the title to "Disruption of macrophage cell volume drives inflammatory responses and type I interferon signalling"

1.2) The authors monitor increased cell volume but never show a picture of the cells in the various experimental conditions. Could they provide such images as additional figures?

Response: We now provide time-lapse images of WT and VRAC KO BMDMs across the experimental conditions in Supplementary Fig 4.

1.3) How does the cell viability change during the experiments?

Response: We have characterised cell viability in WT and VRAC KO macrophages in response to hypo-osmotic media, with additional data now included in figure 5 and figure 7. Whilst no change in viability was detected via an MTT assay (Fig 5A), we observed a slight reduction in cell viability after 6 hours hypo-osmotic shock in WT BMDMs which was enhanced in VRAC KO macrophages assessed by an LDH release assay (Fig 5B). We discovered that this cell swelling-induced cell death was dependent on caspases, as this was prevented by the pan-caspase inhibitor ZVAD, but not by STING inhibition (H151), caspase-1 inhibition (VX765) or necroptosis inhibition (RIPK1 inhibition; necrostatin 1) (Fig 5C). We identified that activity of caspase-3, but not the caspase-1 (typically activated by inflammasomes), was increased by hypo-osmotic shock, and this was also augmented in VRAC-KO macrophages, determined by a caspase-3/7 glo assay (Fig 5D) and immunoblotting for cleaved caspase-3 (Fig 5E). Furthermore, we determine that type I IFN signalling is independent of cell death, since IFNAR KO macrophages still exhibited increases in LDH release and caspase 3 activity in response to hypo-osmotic media (Fig 5F-I). Interestingly, hypo-osmotic cell death was prevented upon co-treatment with PAMPs, particularly TLR7 and TLR9 stimulation (Fig 7I). Together, this demonstrates that whilst not

sufficient to lyse the cells directly, persistent cell swelling induced by hypo-osmotic media induces a delayed activation of programme cell death pathways.

1.4) Are the osmolarity used in the experiments related to known in vivo pathological situations?

Response: We have now amended the discussion to comment on the osmolarities used in this study in relation to in vivo environments. Specifically, we acknowledge that whilst these osmolarities used in vitro are perhaps more severe, our goal was to identify new fundamental concepts on how cell volume influences inflammation. We also highlight the future use of cell-restricted VRAC KO mice as important tools to reveal roles of cell volume control in in vivo pathological situations.

1.5) Figure 3: Are inflammatory cytokines other than CXCL10 modulated by VRAC ? Have the authors tested IL18 as in the in vivo situation?

Response: There are data in the manuscript identifying several inflammatory cytokines other than CXCL10 that are modulated by VRAC. Figure 2 shows modulation of several inflammatory cytokines following exposure to hypo-osmotic media at 4 hours by RNA-seq (Fig 2A-C). Figure 2H shows VRAC modulation of mRNA transcripts of the NF- κ B-driven cytokine IL-1 β at 6 hours. Furthermore, we have now added to this data by measuring transcripts for the inflammatory cytokines *Tnf*, *Ifna* and *Il18* at 6 hours in hypo-osmotic media by qPCR (Supplementary Fig 2A-C).

1.6) Figure 5: The effect of the absence of VRAC on the inflammatory response is relatively minor and only significant for the secretion of IL-18. Is this related to the mode of secretion of this cytokine? Related to pore formation?

Response: This is an interesting point. We have amended the discussion to include this point, referencing that it is intriguing that IL-18 is particularly affected, and propose that this could be due to increased inflammasome activation, IFN signalling or alterations in membrane permeability.

1.7) What dictated the choice of the in vivo experimental set up? Authors might have found a greater role for cell volume in immune responses in the context of a viral infection, where type I IFN plays a more prominent role? This could be better explained in the discussion, at least.

Response: We have amended the results and the discussion to provide more rationale on the choice of in vivo model. We chose a model of systemic hyperinflammation as it produces robust inflammation in tissues. This model is unlike acute inflammation models (i.e. single injection models), since multiple injections of CpG-DNA over 10 days are required, allowing time for establishment of highly inflammatory tissue environments where factors that influence cell volume can develop. Importantly, this model mimics that of hyperinflammatory responses in severe viral infection (PMID:39066167, PMID:34074585) further highlighting its relevancy here.

We agree with the reviewer that our findings are potentially very relevant to contexts with viral infection. Therefore, we now provide data examining cell volume with influenza A infection models. This is added to the manuscript as new figure 8. Interestingly, supporting our data with hypotonic-induced cell swelling, we show that VRAC KO macrophages exhibited enhanced type I IFN and ISG responses following influenza A infection (Fig 8A-D) (notably in isotonic conditions). We found that VRAC-deficiency did not impact viral burden at 24 hours (Fig 8E-F). Moreover, we find that expression of *Lrrc8a*, the essential subunit of VRAC, was downregulated in IAV-infected WT macrophages (Fig 8G). Together, these data indicate the cell volume control could be important in viral sensing and antiviral responses, and therefore using in vivo viral infection models with myeloid-specific VRAC KO mice is an important avenue of future research.

Reviewer #2:

2.1) Figure 1. The comments on the transcriptional data across the conditions are not exhaustive. Before focusing on inflammation, authors should extensively elaborate on the different pathways induced by hypo in WT and in VRAC and in WThypo vs VRACHypo. Interesting genes related to cytoskeletal remodelling and Rho signalling (Arhgap29, Dsg2, Dsg3) are not commented.

Response: *We have now performed further analysis of the RNA-sequencing dataset. In addition to describing our observations on inflammation and immune responses, we have amended the text to elaborate on observed changes in cholesterol synthesis, mRNA processing and cytoskeletal remodelling. This has led to additional investigation into translational arrest pathways shown in new figure 4. We also include additional analysis comparing DEGs involved in these pathways between WT and VRAC KO macrophages in hypotonic conditions. These data are now included in Supplementary Fig 1 and Supplementary Table 1.*

2.2) For data in Figure 2A-D, explain better in methods or legends how are the logFC2 values calculated. The finding that some genes are already slightly elevated in VRAC control should be commented.

Response: *We have amended the methods to include a more detailed description of how Log2FC values were calculated. Log2FC values for each comparison were calculated in DESeq2 using the standard negative binomial model and the apeglm method for shrinkage estimation. For comparisons between hypotonic and isotonic-treated cells within the same genotype, paired analysis was used to correct for within-group expression variability at baseline. The slight elevation in certain genes in VRAC KO control is interesting, but minimal and not significant. The slight elevation in certain genes in VRAC KO control is interesting, but minimal and not significant.*

2.3) In Figure 3, the inhibitors used have controversial and broad effects and are not sufficient to claim or exclude the different pathways. The analysis should be performed in STING or cGAS-depleted macrophages (CRISPR editing is feasible in BMDM) to make a point.

Response: *We have now performed RNAi on cGAS and STING in WT and VRAC KO BMDMs, with data added to Figure 3. Surprisingly, we found that despite >50% knockdown of STING (Fig 3M) and cGAS (Fig 3N), there was no effect on hypo-osmotic media-induced type I IFN induction or ISG production (Fig 3O-P). Following this finding, we have amended the text to conclude that STING alone may not be responsible, with potential off-target effects of H151.*

Following this, we have performed additional mechanistic studies exploring other potential pathways that could be responsible. We provide new data confirming that that hypo-osmotic induction of the ISG viperin is TBK1/IKKε-dependent (blocked by MRT67307, a TBK1/IKKε inhibitor) (Fig 3A-B). We rule out activation of endosomal TLRs using bafilomycin A, an inhibitor of endosomal TLR3, 7, 8 and 9 (PMID: 20346772), which had no impact on hypo-osmotic-induced viperin induction (Fig 3C-D). Thus, we conclude that hypo-osmotic media drives IFNβ signalling through a TBK-1-dependent pathway, independent of endosomal TLRs.

We also provide new data exploring RNA-sensing following mRNA translation inhibition (as revealed by our RNA-seq analysis (Fig 1 + Supplementary Fig 1) as a potential pathway of cell swelling IFN signalling. Added to the manuscript as a new figure 4, we confirmed that hypo-osmotic treatment reduces new protein translation using a puromycin incorporation assay (Fig 4A). Similarly, we observed stress granule formation shortly after hypo-osmotic shock in WT and VRAC KO BMDMs, a readout for build-up of unprocessed RNAs (Fig 4B). Thus, we used RNAi to knockdown MAVS, a common downstream regulator of many RNA-

sensing pathways that induces type I IFNs via TBK1 but observed no difference in *Irfn* or *CXCL10* production following hypo-osmotic treatment in WT or VRAC KO macrophages (Fig 4C-E).

2.4) Second, measuring intracellular cGAMP and transfecting dsDNA into the cytosol would be important to understand if STING hyperactivation in hypo high is ligand-independent.

2.5) More critically: what is the mechanism behind enhanced STING activation upon swelling in VRAC ko? Is it enhanced dimerization, or increased ER to ERGIC translocation or as suggested in the discussion plasmamembrane disruption? The mechanism has to experimentally addressed.

Response (2.4 + 2.5): As we have determined cGAS is not involved, via two chemical compounds (Fig 3H-K) and RNAi (Fig 3N-P), we have not assessed cGAMP levels following hypotonic treatment. The revised manuscript provides extensive additional mechanistic insight into how cell swelling influences STING activation, using the small molecule activator of STING (CMA), added to new figure 6. CMA works directly at STING, downstream and independent of cGAS and 2'3'cGAMP (PMID: 23604073), preventing any involvement of cGAMP transfer in the macrophage response. We demonstrate that STING signalling following CMA treatment is enhanced by cell swelling via *Irfn* induction (Fig 6E), phosphorylation of IRF3 + TBK1 (Fig 6F-H), and STING dimerization (Fig 6I-J), which was augmented further in VRAC KO macrophages. We show that this enhancement is dependent on ER-ERGIC translocation as brefeldin A blocked the response (Fig 6F-H, 6K, 6N). Interestingly, whilst STING signalling and *Irfn* induction was enhanced following CMA treatment in hypo-osmotic media, we observed an absence of IFN β released from VRAC KO macrophages and downstream ISG induction (Fig 6K-N), matching the block in mRNA translation observed (Fig 1, supplementary Fig 1, Fig 4). We propose cell swelling-induced block of translation is reversible as replacement of the hypo-osmotic media with DMEM restored IFN- β release (Fig 6O). Additionally, In support of plasma membrane disruption as a potential mechanism in driving type I IFN, we show that treatment with lipofectamine reagent alone (which causes plasma membrane disruption) was sufficient to promote robust induction of viperin, which was blocked by co-treatment of the plasma membrane stabilising agent punicalagin (Fig 8H).

2.6) Given the previous data on inflammasome, it would be important to dose IL-1b upon STING blockade to understand the cross-talk/selectivity between the pathways.

Response: We have confirmed the inflammasome is not activated, shown by an absence of caspase-1 cleavage in these experimental conditions (Fig 5E) and effect of caspase-1 inhibitors on cell swelling-induced cell death (Fig 5C). To further demonstrate that IL-1 is not influencing the induction of inflammation in these experiments, we provide data to the reviewers here showing that addition of recombinant IL-1 receptor antagonist (IL-1RA) does not prevent hypo-osmotic induction of the ISG viperin (Fig R1).

Figure R1 – IL-1 signalling does not contribute to the ISG response induced by cell swelling. WT and VRAC KO BMDMs were incubated in iso-osmotic or hypo-osmotic media ± recombinant IL-1 receptor antagonist (IL-1ra, 10/100 µg/mL) for 6 h. Lysates were immunoblotted for viperin (n=2).

2.7) In Figure 5: The increased cytokine storm after repeated CpG stimulation in VRAC is interesting. However, the link between macrophage activation and levels of plasma cytokine is very indirect. Data should be provided on macrophages in vivo to show the volume and the expression of some of the cytokines.

Response: This is an extremely interesting, but unfortunately technically challenging point, and something that we are actively working on addressing in our future studies. We now acknowledge this limitation of our study in the discussion. To demonstrate a direct link between macrophage inflammation and VRAC in disease contexts, we now provide data on influenza A infection of WT and VRAC KO macrophages. We show that VRAC KO macrophages directly infected with the RNA virus influenza A have increased type I IFN signalling (Fig 8A-D), linking VRAC to macrophage activation. Importantly, influenza A induces type I IFN responses independent of cGAS (i.e. is still present in cGAS KO macrophages (PMID: 26893169)), demonstrating that this is occurring independent of cGAMP transport.

Reviewer #3:

3.1) The authors use inhibitors to propose that a cGAS-independent pathway is induced by cell volume changes. Is there precedent for this? I would like to see this discussed in the discussion, specifically under what conditions has this been observed in the past and how does it relate to the present observation.

Response: We have now included new text in the discussion to highlight the precedent of cGAS-independent STING activation related to our observations.

This next part would be nice but, I want to be clear, is not required for revisions. Macrophages are notoriously good at pumping drugs/inhibitors out of the cell, is it possible to entertain a genetic model for confirming that hypo-osmotic medium drives a cGAS-independent pathway? I don't want to suggest generating KOs, but I see that Invivogen sells cGAS KO macrophage lines. I'm also not suggesting purchasing them (because as usual they are expensive!) but perhaps a quick e-mail to see if anyone at their present institute has these? Would STING signalling be intact upon exposure of those cells to hypo-osmotic medium?

Response: We now provide data using RNAi against cGAS and STING. Please see response 2.3 above.

3.2) When discussing figure 3K-L, the authors state that the 3 hour time point is selected to limit the contribution of hypo-osmotic medium alone on *Ifnb* expression. This seems at odds the actual data in figure 2D where *Ifnb* expression is increased precisely between 2 and 4 hours. Yet, there is no *Ifnb* response upon exposure to hypo-osmotic medium in VRAC KO cells in figure 3K. What accounts for this apparent inconsistency? Can the authors explain this in the text.

Response: *Interestingly 3 hours is not sufficient to induce IFN β as demonstrated here. The time-course data in 2D does not measure *Ifnb* between 2 and 4 hours so the induction must occur later at a time point later than 3 hours and closer to 4 hours.*

3.3) There is some literature of VRAC/LRRc8 on the regulation of endosome swelling. I wonder if the authors can entertain the idea that loss of VRAC may in some way impact endosomal integrity and hence induce either inflammasome activity or the escape of poly(I:C)/LPS from endosomes. Indeed, macrophages are macropinocytic and constitutively form large endocytic organelles that will likely contain the poly(I:C) and LPS that the cells are bathed in. Loss of endosomal integrity has been linked to inflammasome activity as well. Could this, at least in part, be the basis of the observed synergy of VRAC-deficiency and inflammatory signalling? In an ideal world, it would be nice to see a galectin 3 stain in the VRAC-deficient macrophages that are exposed to hypo-osmotic medium. However, a brief mention of this possibility, if the authors agree that it is logical, in the discussion would suffice. It would help because in the current manuscript there is no mechanism for this synergy.

Response: *This is an excellent idea. To test this, we have now performed galectin-3 staining in WT and VRAC KO BMDMs in hypo-osmotic media. Whilst treatment with LLOME, a lysosomal permeabilising agent, induced galectin-3 puncta formation, hypo-osmotic media did not in WT or VRAC KO BMDMs, suggesting VRAC does not mediate lysosomal permeabilization (Supplementary Fig 5A). To further explore potential lysosomal dysfunction following cell swelling, we also used a DQ-BSA assay. DQ-BSA is self-quenching and becomes fluorescent when processed in the lysosomes. Converse to expectations of reduced lysosomal activity, we found that hypo-osmotic media potentially enhanced DQ-BSA processing, which was not significantly different between WT and VRAC KO BMDMs (Supplementary Fig 5B).*

3.4) Finally, I'm not sure how the in vivo work relates to the "less severe" hypo-osmotic conditions used throughout the paper. It does not really have the ability to discern the degree of cell swelling and in fact the only cytokine that is differentially modulated is IL-18 which is often associated with the NLRP3 inflammasome. So is this more consistent with "severe" swelling. Please discuss.

Response: *We have now amended the discussion to comment on the osmolarities used in this study in relation to in vivo environments. Please see response to point 1.4 above.*

February 13, 2026

RE: JCB Manuscript #202411133R

Jack Green
University of Manchester

Dear Dr. Green:

Thank you for submitting your revised manuscript entitled "Disruption of macrophage cell volume drives inflammatory responses and type I interferon signalling". The paper has now been seen again by the original reviewers and we would be happy to publish your paper in JCB pending final revisions necessary to meet our formatting guidelines (see details below).

****As you will see, reviewer #3 has one lingering concern/comment. In order to address this final issue, we would like for you to do your best to illustrate the caveats of your approach as clearly as possible in the revised manuscript.****

A. MANUSCRIPT ORGANIZATION AND FORMATTING:

1) Text limits: Character count for Articles and Tools is < 40,000, not including spaces. Count includes the abstract, introduction, results, discussion, and acknowledgments. Count does not include title page, materials and methods, figure legends, references, tables, or supplemental legends. At the moment, the manuscript is just below this limit. If you need to exceed the limit slightly in order to address reviewer #3's final comments, that would be fine.

2) Figure formatting: Scale bars must be present on all microscopy images, including inset magnifications. Molecular weight or nucleic acid size markers must be included on all gel electrophoresis.

3) Statistical analysis: Error bars on graphic representations of numerical data must be clearly described in the figure legend. The number of independent data points (n) represented in a graph must be indicated in the legend. Statistical methods should be explained in full in the materials and methods. For figures presenting pooled data the statistical measure should be defined in the figure legends. Please also be sure to indicate the statistical tests used in each of your experiments (both in the figure legend itself and in a separate methods section) as well as the parameters of the test (for example, if you ran a t-test, please indicate if it was one- or two-sided, etc.). Also, if you used parametric tests, please indicate if the data distribution was tested for normality (and if so, how). If not, you must state something to the effect that "Data distribution was assumed to be normal but this was not formally tested."

4) Materials and methods: Should be comprehensive and not simply reference a previous publication for details on how an experiment was performed. Please provide full descriptions (at least in brief) in the text for readers who may not have access to referenced manuscripts. The text should not refer to methods "...as previously described."

5) Please be sure to provide the sequences for all of your primers/oligos and RNAi constructs in the materials and methods. You must also indicate in the methods the source, species, and catalog numbers (where appropriate) for all of your antibodies.

6) Microscope image acquisition: The following information must be provided about the acquisition and processing of images:

- a. Make and model of microscope
- b. Type, magnification, and numerical aperture of the objective lenses
- c. Temperature
- d. imaging medium
- e. Fluorochromes
- f. Camera make and model
- g. Acquisition software
- h. Any software used for image processing subsequent to data acquisition. Please include details and types of operations involved (e.g., type of deconvolution, 3D reconstitutions, surface or volume rendering, gamma adjustments, etc.).

7) References: There is no limit to the number of references cited in a manuscript. References should be cited parenthetically in the text by author and year of publication. Abbreviate the names of journals according to PubMed.

8) Supplemental materials: There are strict limits on the allowable amount of supplemental data. Articles/Tools may have up to 5

supplemental figures. At the moment, you meet this limit but please bear it in mind when revising.

Please also note that tables, like figures, should be provided as individual, editable files. A summary of all supplemental material (that is, in addition to the supplementary figure legends) should appear at the end of the Materials and methods section. Please see any recent JCB paper for an example of this.

9) Conflict of interest statement: JCB requires inclusion of a statement in the acknowledgements regarding competing financial interests. If no competing financial interests exist, please include the following statement: "The authors declare no competing financial interests." If competing interests are declared, please follow your statement of these competing interests with the following statement: "The authors declare no further competing financial interests."

10) A separate author contribution section is required following the Acknowledgments in all research manuscripts. All authors should be mentioned and designated by their first and middle initials and full surnames. We encourage use of the CRediT nomenclature (<https://casrai.org/credit/>).

11) ORCID IDs: ORCID IDs are unique identifiers allowing researchers to create a record of their various scholarly contributions in a single place. Please note that ORCID IDs are now *required* for all authors. At resubmission of your final files, please be sure to provide your ORCID ID and those of all co-authors.

12) Journal of Cell Biology now requires a data availability statement for all research article submissions. These statements will be published in the article directly above the Acknowledgments. The statement should address all data underlying the research presented in the manuscript. Please visit the JCB instructions for authors for guidelines and examples of statements at (<https://rupress.org/jcb/pages/editorial-policies#data-availability-statement>).

B. FINAL FILES:

Thank you for your attention to these final processing requirements. Please revise and format the manuscript and upload materials within 7-14 days. If you need an extension for whatever reason, please let us know and we can work with you to determine a suitable revision period.

Thank you for this interesting contribution, we look forward to publishing your paper in Journal of Cell Biology.

Sincerely,

Ana-María Lennon-Dumenil, PhD
Monitoring Editor

Tim Spencer, PhD
Executive Editor
Journal of Cell Biology

Reviewer #1 (Comments to the Authors (Required)):

The authors provide evidence supporting the role of VRAC in the regulation of cell volume in macrophages in hypoosmotic conditions and how a dysfunction of VRAC and sustained increase in cell volume induces inflammatory pathways. With this revised version, the authors responded carefully to all the comments made to them. The numerous additional results provided in this version of the manuscript really strengthen the data, in particular concerning the signaling pathways and the potential antiviral effect associated with sustained cell swelling.

Reviewer #2 (Comments to the Authors (Required)):

The revised version of the manuscript has addressed my main points and new important data now clarify the mechanism of inflammatory response upon cell swelling. I also appreciate the caution in interpreting the in vivo cytokine storm results.

Reviewer #3 (Comments to the Authors (Required)):

The authors have been very responsive to the comments from the initial round of review. One concern though is that these revisions have made the findings less conclusive. For example, the new RNAi experiments do not necessarily agree with or strengthen the previous conclusions drawn from inhibitor experiments. I commend the authors for adding additional experiments on TBK and TLRs, but we are left with the same issue as the original round - mechanisms based solely on drug inhibitions. In sum, I suggest publication but do feel that there are still some gaps in the mechanism.